# A Normative Theory of Adaptive Dimensionality Reduction in Neural Networks

**Cengiz Pehlevan**
Simons Center for Data Analysis
Simons Foundation
New York, NY 10010
cpehlevan@simonsfoundation.org

**Dmitri B. Chklovskii**
Simons Center for Data Analysis
Simons Foundation
New York, NY 10010
dchklovskii@simonsfoundation.org

## Abstract

To make sense of the world our brains must analyze high-dimensional datasets streamed by our sensory organs. Because such analysis begins with dimensionality reduction, modeling early sensory processing requires biologically plausible online dimensionality reduction algorithms. Recently, we derived such an algorithm, termed similarity matching, from a Multidimensional Scaling (MDS) objective function. However, in the existing algorithm, the number of output dimensions is set a priori by the number of output neurons and cannot be changed. Because the number of informative dimensions in sensory inputs is variable there is a need for adaptive dimensionality reduction. Here, we derive biologically plausible dimensionality reduction algorithms which adapt the number of output dimensions to the eigenspectrum of the input covariance matrix. We formulate three objective functions which, in the offline setting, are optimized by the projections of the input dataset onto its principal subspace scaled by the eigenvalues of the output covariance matrix. In turn, the output eigenvalues are computed as i) soft-thresholded, ii) hard-thresholded, iii) equalized thresholded eigenvalues of the input covariance matrix. In the online setting, we derive the three corresponding adaptive algorithms and map them onto the dynamics of neuronal activity in networks with biologically plausible local learning rules. Remarkably, in the last two networks, neurons are divided into two classes which we identify with principal neurons and interneurons in biological circuits.

## 1 Introduction

Our brains analyze high-dimensional datasets streamed by our sensory organs with efficiency and speed rivaling modern computers. At the early stage of such analysis, the dimensionality of sensory inputs is drastically reduced as evidenced by anatomical measurements. Human retina, for example, conveys signals from $\approx$125 million photoreceptors to the rest of the brain via $\approx$1 million ganglion cells [1] suggesting a hundred-fold dimensionality reduction. Therefore, biologically plausible dimensionality reduction algorithms may offer a model of early sensory processing.

In a seminal work [2] Oja proposed that a *single* neuron may compute the *first* principal component of activity in upstream neurons. At each time point, Oja's neuron projects a vector composed of firing rates of upstream neurons onto the vector of synaptic weights by summing up currents generated by its synapses. In turn, synaptic weights are adjusted according to a Hebbian rule depending on the activities of only the postsynaptic and corresponding presynaptic neurons [2].

Following Oja's work, many *multineuron* circuits were proposed to extract *multiple* principal components of the input, for a review see [3]. However, most multineuron algorithms did not meet the same level of rigor and biological plausibility as the single-neuron algorithm [2, 4] which can be derived using a normative approach, from a principled objective function [5], and contains only lo-

cal Hebbian learning rules. Algorithms derived from principled objective functions either did not possess local learning rules [6, 4, 7, 8] or had other biologically implausible features [9]. In other algorithms, local rules were chosen heuristically rather than derived from a principled objective function [10, 11, 12, 9, 3, 13, 14, 15, 16].

There is a notable exception to the above observation but it has other shortcomings. The two-layer circuit with reciprocal synapses [17, 18, 19] can be derived from the minimization of the representation error. However, the activity of principal neurons in the circuit is a dummy variable without its own dynamics. Therefore, such principal neurons do not integrate their input in time, contradicting existing experimental observations.

Other normative approaches use an information theoretical objective to compare theoretical limits with experimentally measured information in single neurons or populations [20, 21, 22] or to calculate optimal synaptic weights in a postulated neural network [23, 22].

Recently, a novel approach to the problem has been proposed [24]. Starting with the Multidimensional Scaling (MDS) strain cost function [25, 26] we derived an algorithm which maps onto a neuronal circuit with local learning rules. However, [24] had major limitations, which are shared by vairous other multineuron algorithms:

1. The number of output dimensions was determined by the fixed number of output neurons precluding adaptation to the varying number of informative components. A better solution would be to let the network decide, depending on the input statistics, how many dimensions to represent [14, 15]. The dimensionality of neural activity in such a network would be usually less than the maximum set by the number of neurons.
2. Because output neurons were coupled by anti-Hebbian synapses which are most naturally implemented by inhibitory synapses, if these neurons were to have excitatory outputs, as suggested by cortical anatomy, they would violate Dale's law (i.e. each neuron uses only one fast neurotransmitter). Here, following [10], by anti-Hebbian we mean synaptic weights that get more negative with correlated activity of pre- and postsynaptic neurons.
3. The output had a wide dynamic range which is difficult to implement using biological neurons with a limited range. A better solution [27, 13] is to equalize the output variance across neurons.

In this paper, we advance the normative approach of [24] by proposing three new objective functions which allow us to overcome the above limitations. We optimize these objective functions by proceeding as follows. In Section 2, we formulate and solve three optimization problems of the form:

$$\text{Offline setting}: \mathbf{Y}^* = \arg\min_{\mathbf{Y}} L(\mathbf{X}, \mathbf{Y}). \tag{1}$$

Here, the input to the network, $\mathbf{X} = [\mathbf{x}_1, \ldots, \mathbf{x}_T]$ is an $n \times T$ matrix with $T$ centered input data samples in $\mathbb{R}^n$ as its columns and the output of the network, $\mathbf{Y} = [\mathbf{y}_1, \ldots, \mathbf{y}_T]$ is a $k \times T$ matrix with corresponding outputs in $\mathbb{R}^k$ as its columns. We assume $T >> k$ and $T >> n$. Such optimization problems are posed in the so-called offline setting where outputs are computed after seeing all data.

Whereas the optimization problems in the offline setting admit closed-form solution, such setting is ill-suited for modeling neural computation on the mechanistic level and must be replaced by the online setting. Indeed, neurons compute an output, $\mathbf{y}_T$, for each data sample presentation, $\mathbf{x}_T$, before the next data sample is presented and past outputs cannot be altered. In such online setting, optimization is performed at every time step, $T$, on the objective which is a function of all inputs and outputs up to time $T$. Moreover, an online algorithm (also known as streaming) is not capable of storing all previous inputs and outputs and must rely on a smaller number of state variables.

In Section 3, we formulate three corresponding online optimization problems with respect to $\mathbf{y}_T$, while keeping all the previous outputs fixed:

$$\text{Online setting}: \mathbf{y}_T \leftarrow \arg\min_{\mathbf{y}_T} L(\mathbf{X}, \mathbf{Y}). \tag{2}$$

Then we derive algorithms solving these problems online and map their steps onto the dynamics of neuronal activity and local learning rules for synaptic weights in three neural networks.

We show that the solutions of the optimization problems and the corresponding online algorithms remove the limitations outlined above by performing the following computational tasks:

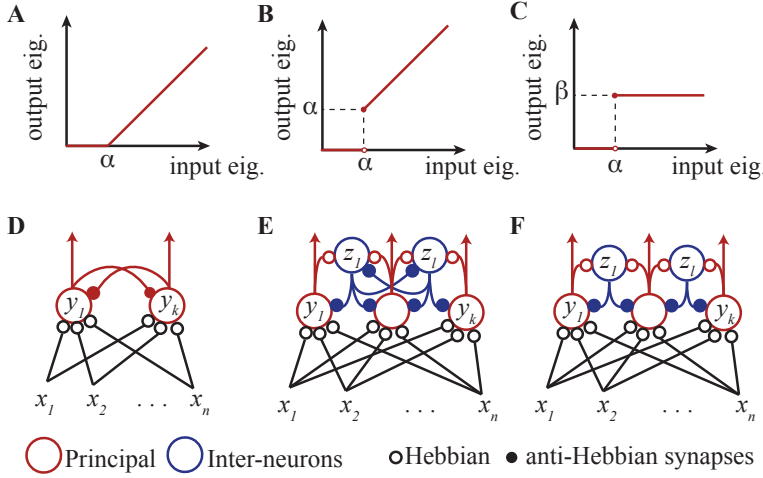

Figure 1: Input-output functions of the three offline solutions and neural network implementations of the corresponding online algorithms. **A-C.** Input-output functions of covariance eigenvalues. **A.** Soft-thresholding. **B.** Hard-thresholding. **C.** Equalization after thresholding. **D-F.** Corresponding network architectures.

1. Soft-thresholding the eigenvalues of the input covariance matrix, Figure 1A: eigenvalues below the threshold are set to zero and the rest are shrunk by the threshold magnitude. Thus, the number of output dimensions is chosen adaptively. This algorithm maps onto a single-layer neural network with the same architecture as in [24], Figure 1D, but with modified learning rules.
2. Hard-thresholding of input eigenvalues, Figure 1B: eigenvalues below the threshold vanish as before, but eigenvalues above the threshold remain unchanged. The steps of such algorithm map onto the dynamics of neuronal activity in a network which, in addition to principal neurons, has a layer of interneurons reciprocally connected with principal neurons and each other, Figure 1E.
3. Equalization of non-zero eigenvalues, Figure 1C. The corresponding network's architecture, Figure 1F, lacks reciprocal connections among interneurons. As before, the number of above-threshold eigenvalues is chosen adaptively and cannot exceed the number of principal neurons. If the two are equal, this network whitens the output.

In Section 4, we demonstrate that the online algorithms perform well on a synthetic dataset and, in Discussion, we compare our neural circuits with biological observations.

## 2 Dimensionality reduction in the offline setting

In this Section, we introduce and solve, in the offline setting, three novel optimization problems whose solutions reduce the dimensionality of the input. We state our results in three Theorems which are proved in the Supplementary Material.

### 2.1 Soft-thresholding of covariance eigenvalues

We consider the following optimization problem in the offline setting:

$$\min_{\mathbf{Y}} \left\| \mathbf{X}^\top \mathbf{X} - \mathbf{Y}^\top \mathbf{Y} - \alpha T \mathbf{I}_T \right\|_F^2, \tag{3}$$

where $\alpha \geq 0$ and $\mathbf{I}_T$ is the $T \times T$ identity matrix. To gain intuition behind this choice of the objective function let us expand the squared norm and keep only the $\mathbf{Y}$-dependent terms:

$$\arg\min_{\mathbf{Y}} \left\| \mathbf{X}^\top \mathbf{X} - \mathbf{Y}^\top \mathbf{Y} - \alpha T \mathbf{I}_T \right\|_F^2 = \arg\min_{\mathbf{Y}} \left\| \mathbf{X}^\top \mathbf{X} - \mathbf{Y}^\top \mathbf{Y} \right\|_F^2 + 2\alpha T \operatorname{Tr}\left( \mathbf{Y}^\top \mathbf{Y} \right), \tag{4}$$

where the first term matches the similarity of input and output[24] and the second term is a nuclear norm of $\mathbf{Y}^\top \mathbf{Y}$ known to be a convex relaxation of the matrix rank used for low-rank matrix modeling [28]. Thus, objective function (3) enforces low-rank similarity matching.

We show that the optimal output $\mathbf{Y}$ is a projection of the input data, $\mathbf{X}$, onto its principal subspace. The subspace dimensionality is set by $m$, the number of eigenvalues of the data covariance matrix, $\mathbf{C} = \frac{1}{T}\mathbf{X}\mathbf{X}^\top = \frac{1}{T}\sum_{t=1}^{T} \mathbf{x}_t \mathbf{x}_t^\top$, that are greater than or equal to the parameter $\alpha$.

**Theorem 1.** *Suppose an eigen-decomposition of* $\mathbf{X}^\top \mathbf{X} = \mathbf{V}^X \mathbf{\Lambda}^X \mathbf{V}^{X\top}$, *where* $\mathbf{\Lambda}^X = \mathrm{diag}\left(\lambda_1^X, \ldots, \lambda_T^X\right)$ *with* $\lambda_1^X \geq \ldots \geq \lambda_T^X$. *Note that* $\mathbf{\Lambda}^X$ *has at most* $n$ *nonzero eigenvalues coinciding with those of* $T\mathbf{C}$. *Then,*

$$\mathbf{Y}^* = \mathbf{U}_k \,\mathbf{ST}_k(\mathbf{\Lambda}^X, \alpha T)^{1/2}\, \mathbf{V}_k^{X\top}, \tag{5}$$

*are optima of* (3), *where* $\mathbf{ST}_k(\mathbf{\Lambda}^X, \alpha T) = \mathrm{diag}\left(\mathrm{ST}\left(\lambda_1^X, \alpha T\right), \ldots, \mathrm{ST}\left(\lambda_k^X, \alpha T\right)\right)$, $\mathrm{ST}$ *is the soft-thresholding function,* $\mathrm{ST}(a, b) = \max(a - b, 0)$, $\mathbf{V}_k^X$ *consists of the columns of* $\mathbf{V}^X$ *corresponding to the top* $k$ *eigenvalues, i.e.* $\mathbf{V}_k^X = \left[\mathbf{v}_1^X, \ldots, \mathbf{v}_k^X\right]$ *and* $\mathbf{U}_k$ *is any* $k \times k$ *orthogonal matrix, i.e.* $\mathbf{U}_k \in O(k)$. *The form* (5) *uniquely defines all optima of* (3), *except when* $k < m$, $\lambda_k^X > \alpha T$ *and* $\lambda_k^X = \lambda_{k+1}^X$.

## 2.2 Hard-thresholding of covariance eigenvalues

Consider the following minimax problem in the offline setting:

$$\min_{\mathbf{Y}} \max_{\mathbf{Z}} \left\|\mathbf{X}^\top \mathbf{X} - \mathbf{Y}^\top \mathbf{Y}\right\|_F^2 - \left\|\mathbf{Y}^\top \mathbf{Y} - \mathbf{Z}^\top \mathbf{Z} - \alpha T \mathbf{I}_T\right\|_F^2, \tag{6}$$

where $\alpha \geq 0$ and we introduced an internal variable $\mathbf{Z}$, which is an $l \times T$ matrix $\mathbf{Z} = [\mathbf{z}_1, \ldots, \mathbf{z}_T]$ with $\mathbf{z}_t \in \mathbb{R}^l$. The intuition behind this objective function is again based on similarity matching but rank regularization is applied indirectly via the internal variable, $\mathbf{Z}$.

**Theorem 2.** *Suppose an eigen-decomposition of* $\mathbf{X}^\top \mathbf{X} = \mathbf{V}^X \mathbf{\Lambda}^X \mathbf{V}^{X\top}$, *where* $\mathbf{\Lambda}^X = \mathrm{diag}\left(\lambda_1^X, \ldots, \lambda_T^X\right)$ *with* $\lambda_1^X \geq \ldots \geq \lambda_T^X \geq 0$. *Assume* $l \geq \min(k, m)$. *Then,*

$$\mathbf{Y}^* = \mathbf{U}_k \,\mathbf{HT}_k(\mathbf{\Lambda}^X, \alpha T)^{1/2}\, \mathbf{V}_k^{X\top}, \qquad \mathbf{Z}^* = \mathbf{U}_l \,\mathbf{ST}_{l,\min(k,m)}(\mathbf{\Lambda}^X, \alpha T)^{1/2}\, \mathbf{V}_l^{X\top}, \tag{7}$$

*are optima of* (6), *where* $\mathbf{HT}_k(\mathbf{\Lambda}^X, \alpha T) = \mathrm{diag}\left(\mathrm{HT}\left(\lambda_1^X, \alpha T\right), \ldots, \mathrm{HT}\left(\lambda_k^X, \alpha T\right)\right)$, $\mathrm{HT}(a, b) = a\Theta(a - b)$ *with* $\Theta()$ *being the step function:* $\Theta(a - b) = 1$ *if* $a \geq b$ *and* $\Theta(a - b) = 0$ *if* $a < b$, $\mathbf{ST}_{l,\min(k,m)}(\mathbf{\Lambda}^X, \alpha T) = \mathrm{diag}\big(\mathrm{ST}\left(\lambda_1^X, \alpha T\right), \ldots, \mathrm{ST}\left(\lambda_{\min(k,m)}^X, \alpha T\right) \underbrace{, 0, \ldots, 0}_{l-\min(k,m)}\big), \mathbf{V}_p^X =$

$\left[\mathbf{v}_1^X, \ldots, \mathbf{v}_p^X\right]$ *and* $\mathbf{U}_p \in O(p)$. *The form* (7) *uniquely defines all optima* (6) *except when either 1)* $\alpha$ *is an eigenvalue of* $\mathbf{C}$ *or 2)* $k < m$ *and* $\lambda_k^X = \lambda_{k+1}^X$.

## 2.3 Equalizing thresholded covariance eigenvalues

Consider the following minimax problem in the offline setting:

$$\min_{\mathbf{Y}} \max_{\mathbf{Z}} \mathrm{Tr}\left(-\mathbf{X}^\top \mathbf{X} \mathbf{Y}^\top \mathbf{Y} + \mathbf{Y}^\top \mathbf{Y} \mathbf{Z}^\top \mathbf{Z} + \alpha T \mathbf{Y}^\top \mathbf{Y} - \beta T \mathbf{Z}^\top \mathbf{Z}\right), \tag{8}$$

where $\alpha \geq 0$ and $\beta > 0$. This objective function follows from (6) after dropping the quartic $\mathbf{Z}$ term.

**Theorem 3.** *Suppose an eigen-decomposition of* $\mathbf{X}^\top \mathbf{X}$ *is* $\mathbf{X}^\top \mathbf{X} = \mathbf{V}^X \mathbf{\Lambda}^X \mathbf{V}^{X\top}$, *where* $\mathbf{\Lambda}^X = \mathrm{diag}\left(\lambda_1^X, \ldots, \lambda_T^X\right)$ *with* $\lambda_1^X \geq \ldots \geq \lambda_T^X \geq 0$. *Assume* $l \geq \min(k, m)$. *Then,*

$$\mathbf{Y}^* = \mathbf{U}_k \,\sqrt{\beta T}\,\mathbf{\Theta}_k(\mathbf{\Lambda}^X, \alpha T)^{1/2}\, \mathbf{V}_k^{X\top}, \qquad \mathbf{Z}^* = \mathbf{U}_l \,\mathbf{\Sigma}_{l \times T}\mathbf{O}_{\mathbf{\Lambda}^{Y*}}\mathbf{V}^{X\top}, \tag{9}$$

*are optima of* (8), *where* $\mathbf{\Theta}_k(\mathbf{\Lambda}^X, \alpha T) = \mathrm{diag}\left(\Theta\left(\lambda_1^X - \alpha T\right), \ldots, \Theta\left(\lambda_k^X - \alpha T\right)\right)$, $\mathbf{\Sigma}_{l \times T}$ *is an* $l \times T$ *rectangular diagonal matrix with top* $\min(k, m)$ *diagonals are set to arbitrary nonnegative constants and the rest are zero,* $\mathbf{O}_{\mathbf{\Lambda}^{Y*}}$ *is a block-diagonal orthogonal matrix that has two blocks: the top block is* $\min(k, m)$ *dimensional and the bottom block is* $T - \min(k, m)$ *dimensional,* $\mathbf{V}_p = \left[\mathbf{v}_1^X, \ldots, \mathbf{v}_p^X\right]$, *and* $\mathbf{U}_p \in O(p)$. *The form* (9) *uniquely defines all optima of* (8) *except when either 1)* $\alpha$ *is an eigenvalue of* $\mathbf{C}$ *or 2)* $k < m$ *and* $\lambda_k^X = \lambda_{k+1}^X$.

*Remark* 1. If $k = m$, then $\mathbf{Y}$ is full-rank and $\frac{1}{T}\mathbf{Y}\mathbf{Y}^\top = \beta \mathbf{I}_k$, implying that the output is whitened, equalizing variance across all channels.

# 3  Online dimensionality reduction using Hebbian/anti-Hebbian neural nets

In this Section, we formulate online versions of the dimensionality reduction optimization problems presented in the previous Section, derive corresponding online algorithms and map them onto the dynamics of neural networks with biologically plausible local learning rules. The order of subsections corresponds to that in the previous Section.

## 3.1 Online soft-thresholding of eigenvalues

Consider the following optimization problem in the online setting:

$$\mathbf{y}_T \leftarrow \underset{\mathbf{y}_T}{\arg\min} \left\| \mathbf{X}^\top \mathbf{X} - \mathbf{Y}^\top \mathbf{Y} - \alpha T \mathbf{I}_T \right\|_F^2. \tag{10}$$

By keeping only the terms that depend on $\mathbf{y}_T$ we get the following objective for (2):

$$L = -4\mathbf{x}_T^\top \left( \sum_{t=1}^{T-1} \mathbf{x}_t \mathbf{y}_t^\top \right) \mathbf{y}_T + 2\mathbf{y}_T^\top \left( \sum_{t=1}^{T-1} \mathbf{y}_t \mathbf{y}_t^\top + \alpha T \mathbf{I}_m \right) \mathbf{y}_T - 2\|\mathbf{x}_T\|^2 \|\mathbf{y}_T\|^2 + \|\mathbf{y}_T\|^4. \tag{11}$$

In the large-$T$ limit, the last two terms can be dropped since the first two terms grow linearly with $T$ and dominate. The remaining cost is a positive definite quadratic form in $\mathbf{y}_T$ and the optimization problem is convex. At its minimum, the following equality holds:

$$\left( \sum_{t=1}^{T-1} \mathbf{y}_t \mathbf{y}_t^\top + \alpha T \mathbf{I}_m \right) \mathbf{y}_T = \left( \sum_{t=1}^{T-1} \mathbf{y}_t \mathbf{x}_t^\top \right) \mathbf{x}_T. \tag{12}$$

While a closed-form analytical solution via matrix inversion exists for $\mathbf{y}_T$, we are interested in biologically plausible algorithms. Instead, we use a weighted Jacobi iteration where $\mathbf{y}_T$ is updated according to:

$$\mathbf{y}_T \leftarrow (1 - \eta)\, \mathbf{y}_T + \eta \left( \mathbf{W}_T^{YX} \mathbf{x}_T - \mathbf{W}_T^{YY} \mathbf{y}_T \right), \tag{13}$$

where $\eta$ is the weight parameter, and $\mathbf{W}_T^{YX}$ and $\mathbf{W}_T^{YY}$ are normalized input-output and output-output covariances,

$$W_{T,ik}^{YX} = \frac{\sum_{t=1}^{T-1} y_{t,i} x_{t,k}}{\alpha T + \sum_{t=1}^{T-1} y_{t,i}^2}, \qquad W_{T,i,j\neq i}^{YY} = \frac{\sum_{t=1}^{T-1} y_{t,i} y_{t,j}}{\alpha T + \sum_{t=1}^{T-1} y_{t,i}^2}, \qquad W_{T,ii}^{YY} = 0. \tag{14}$$

Iteration (13) can be implemented by the dynamics of neuronal activity in a single-layer network, Figure 1D. Then, $\mathbf{W}_T^{YX}$ and $\mathbf{W}_T^{YY}$ represent the weights of feedforward ($\mathbf{x}_t \rightarrow \mathbf{y}_t$) and lateral ($\mathbf{y}_t \rightarrow \mathbf{y}_t$) synaptic connections, respectively. Remarkably, synaptic weights appear in the online solution despite their absence in the optimization problem formulation (3). Previously, nonnormalized covariances have been used as state variables in an online dictionary learning algorithm [29].

To formulate a fully online algorithm, we rewrite (14) in a recursive form. This requires introducing a scalar variable $D_{T,i}^Y$ representing cumulative activity of a neuron $i$ up to time $T-1$, $D_{T,i}^Y = \alpha T + \sum_{t=1}^{T-1} y_{t,i}^2$. Then, at each data sample presentation, $T$, after the output $\mathbf{y}_T$ converges to a steady state, the following updates are performed:

$$D_{T+1,i}^Y \leftarrow D_{T,i}^Y + \alpha + y_{T,i}^2,$$
$$W_{T+1,ij}^{YX} \leftarrow W_{T,ij}^{YX} + \left( y_{T,i} x_{T,j} - \left( \alpha + y_{T,i}^2 \right) W_{T,ij}^{YX} \right) / D_{T+1,i}^Y,$$
$$W_{T+1,i,j\neq i}^{YY} \leftarrow W_{T,ij}^{YY} + \left( y_{T,i} y_{T,j} - \left( \alpha + y_{T,i}^2 \right) W_{T,ij}^{YY} \right) / D_{T+1,i}^Y. \tag{15}$$

Hence, we arrive at a neural network algorithm that solves the optimization problem (10) for streaming data by alternating between two phases. After a data sample is presented at time $T$, in the first phase of the algorithm (13), neuron activities are updated until convergence to a fixed point. In the second phase of the algorithm, synaptic weights are updated for feedforward connections according to a local Hebbian rule (15) and for lateral connections according to a local anti-Hebbian rule (due to the $(-)$ sign in equation (13)). Interestingly, in the $\alpha = 0$ limit, these updates have the same form as the single-neuron Oja rule [24, 2], except that the learning rate is not a free parameter but is determined by the cumulative neuronal activity $1/D_{T+1,i}^Y$ [4, 5].

## 3.2 Online hard-thresholding of eigenvalues

Consider the following minimax problem in the online setting, where we assume $\alpha > 0$:

$$\{\mathbf{y}_T, \mathbf{z}_T\} \leftarrow \underset{\mathbf{y}_T}{\arg\min} \, \underset{\mathbf{z}_T}{\arg\max} \left\| \mathbf{X}^\top \mathbf{X} - \mathbf{Y}^\top \mathbf{Y} \right\|_F^2 - \left\| \mathbf{Y}^\top \mathbf{Y} - \mathbf{Z}^\top \mathbf{Z} - \alpha T \mathbf{I}_T \right\|_F^2. \tag{16}$$

By keeping only those terms that depend on $\mathbf{y}_T$ or $\mathbf{z}_T$ and considering the large-$T$ limit, we get the

following objective:

$$L = 2\alpha T \left\| \mathbf{y}_T \right\|^2 - 4\mathbf{x}_T^\top \left( \sum_{t=1}^{T-1} \mathbf{x}_t \mathbf{y}_t^\top \right) \mathbf{y}_T - 2\mathbf{z}_T^\top \left( \sum_{t=1}^{T-1} \mathbf{z}_t \mathbf{z}_t^\top + \alpha T \mathbf{I}_k \right) \mathbf{z}_T + 4\mathbf{y}_T^\top \left( \sum_{t=1}^{T-1} \mathbf{y}_t \mathbf{z}_t^\top \right) \mathbf{z}_T.$$

$$(17)$$

Note that this objective is strongly convex in $\mathbf{y}_T$ and strongly concave in $\mathbf{z}_T$. The solution of this minimax problem is the saddle-point of the objective function, which is found by setting the gradient of the objective with respect to $\{\mathbf{y}_T, \mathbf{z}_T\}$ to zero [30]:

$$\alpha T \mathbf{y}_T = \left( \sum_{t=1}^{T-1} \mathbf{y}_t \mathbf{x}_t^\top \right) \mathbf{x}_T - \left( \sum_{t=1}^{T-1} \mathbf{y}_t \mathbf{z}_t^\top \right) \mathbf{z}_T, \quad \left( \sum_{t=1}^{T-1} \mathbf{z}_t \mathbf{z}_t^\top + \alpha T \mathbf{I}_k \right) \mathbf{z}_T = \left( \sum_{t=1}^{T-1} \mathbf{z}_t \mathbf{y}_t^\top \right) \mathbf{y}_T.$$

$$(18)$$

To obtain a neurally plausible algorithm, we solve these equations by a weighted Jacobi iteration:

$$\mathbf{y}_T \leftarrow (1 - \eta) \mathbf{y}_T + \eta \left( \mathbf{W}_T^{YX} \mathbf{x}_T - \mathbf{W}_T^{YZ} \mathbf{z}_T \right), \quad \mathbf{z}_T \leftarrow (1 - \eta) \mathbf{z}_T + \eta \left( \mathbf{W}_T^{ZY} \mathbf{y}_T - \mathbf{W}_T^{ZZ} \mathbf{z}_T \right).$$

$$(19)$$

Here, similarly to (14), $\mathbf{W}_T$ are normalized covariances that can be updated recursively:

$$D_{T+1,i}^Y \leftarrow D_{T,i}^Y + \alpha, \qquad D_{T+1,i}^Z \leftarrow D_{T,i}^Z + \alpha + z_{T,i}^2$$

$$W_{T+1,ij}^{YX} \leftarrow W_{T,ij}^{YX} + \left( y_{T,i} x_{T,j} - \alpha W_{T,ij}^{YX} \right) / D_{T+1,i}^Y$$

$$W_{T+1,ij}^{YZ} \leftarrow W_{T,ij}^{YZ} + \left( y_{T,i} z_{T,j} - \alpha W_{T,ij}^{YZ} \right) / D_{T+1,i}^Y$$

$$W_{T+1,i,j}^{ZY} \leftarrow W_{T,ij}^{ZY} + \left( z_{T,i} y_{T,j} - \left( \alpha + z_{T,i}^2 \right) W_{T,ij}^{ZY} \right) / D_{T+1,i}^Z$$

$$W_{T+1,i,j \neq i}^{ZZ} \leftarrow W_{T,ij}^{ZZ} + \left( z_{T,i} z_{T,j} - \left( \alpha + z_{T,i}^2 \right) W_{T,ij}^{ZZ} \right) / D_{T+1,i}^Z, \quad W_{T,ii}^{ZZ} = 0. \qquad (20)$$

Equations (19) and (20) define an online algorithm that can be naturally implemented by a neural network with two populations of neurons: principal and interneurons, Figure 1E. Again, after each data sample presentation, $T$, the algorithm proceeds in two phases. First, (19) is iterated until convergence by the dynamics of neuronal activities. Second, synaptic weights are updated according to local, anti-Hebbian (for synapses from interneurons) and Hebbian (for all other synapses) rules.

### 3.3 Online thresholding and equalization of eigenvalues

Consider the following minimax problem in the online setting, where we assume $\alpha > 0$ and $\beta > 0$:

$$\{\mathbf{y}_T, \mathbf{z}_T\} \leftarrow \arg\min_{\mathbf{y}_T} \arg\max_{\mathbf{z}_T} \operatorname{Tr} \left[ -\mathbf{X}^\top \mathbf{X} \mathbf{Y}^\top \mathbf{Y} + \mathbf{Y}^\top \mathbf{Y} \mathbf{Z}^\top \mathbf{Z} + \alpha T \mathbf{Y}^\top \mathbf{Y} - \beta T \mathbf{Z}^\top \mathbf{Z} \right]. \quad (21)$$

By keeping only those terms that depend on $\mathbf{y}_T$ or $\mathbf{z}_T$ and considering the large-$T$ limit, we get the following objective:

$$L = \alpha T \left\| \mathbf{y}_T \right\|^2 - 2\mathbf{x}_T^\top \left( \sum_{t=1}^{T-1} \mathbf{x}_t \mathbf{y}_t^\top \right) \mathbf{y}_T - \beta T \left\| \mathbf{z}_T \right\|^2 + 2\mathbf{y}_T^\top \left( \sum_{t=1}^{T-1} \mathbf{y}_t \mathbf{z}_t^\top \right) \mathbf{z}_T. \qquad (22)$$

This objective is strongly convex in $\mathbf{y}_T$ and strongly concave in $\mathbf{z}_T$ and its saddle point is given by:

$$\alpha T \mathbf{y}_T = \left( \sum_{t=1}^{T-1} \mathbf{y}_t \mathbf{x}_t^\top \right) \mathbf{x}_T - \left( \sum_{t=1}^{T-1} \mathbf{y}_t \mathbf{z}_t^\top \right) \mathbf{z}_T, \qquad \beta T \mathbf{z}_T = \left( \sum_{t=1}^{T-1} \mathbf{z}_t \mathbf{y}_t^\top \right) \mathbf{y}_T. \qquad (23)$$

To obtain a neurally plausible algorithm, we solve these equations by a weighted Jacobi iteration:

$$\mathbf{y}_T \leftarrow (1 - \eta) \mathbf{y}_T + \eta \left( \mathbf{W}_T^{YX} \mathbf{x}_T - \mathbf{W}_T^{YZ} \mathbf{z}_T \right), \qquad \mathbf{z}_T \leftarrow (1 - \eta) \mathbf{z}_T + \eta \mathbf{W}_T^{ZY} \mathbf{y}_T, \qquad (24)$$

As before, $\mathbf{W}_T$ are normalized covariances which can be updated recursively:

$$D_{T+1,i}^Y \leftarrow D_{T,i}^Y + \alpha, \qquad D_{T+1,i}^Z \leftarrow D_{T,i}^Z + \beta$$

$$W_{T+1,ij}^{YX} \leftarrow W_{T,ij}^{YX} + \left( y_{T,i} x_{T,j} - \alpha W_{T,ij}^{YX} \right) / D_{T+1,i}^Y$$

$$W_{T+1,ij}^{YZ} \leftarrow W_{T,ij}^{YZ} + \left( y_{T,i} z_{T,j} - \alpha W_{T,ij}^{YZ} \right) / D_{T+1,i}^Y$$

$$W_{T+1,i,j}^{ZY} \leftarrow W_{T,ij}^{ZY} + \left( z_{T,i} y_{T,j} - \beta W_{T,ij}^{ZY} \right) / D_{T+1,i}^Z. \qquad (25)$$

Equations (24) and (25) define an online algorithm that can be naturally implemented by a neural network with principal neurons and interneurons. As beofre, after each data sample presentation at

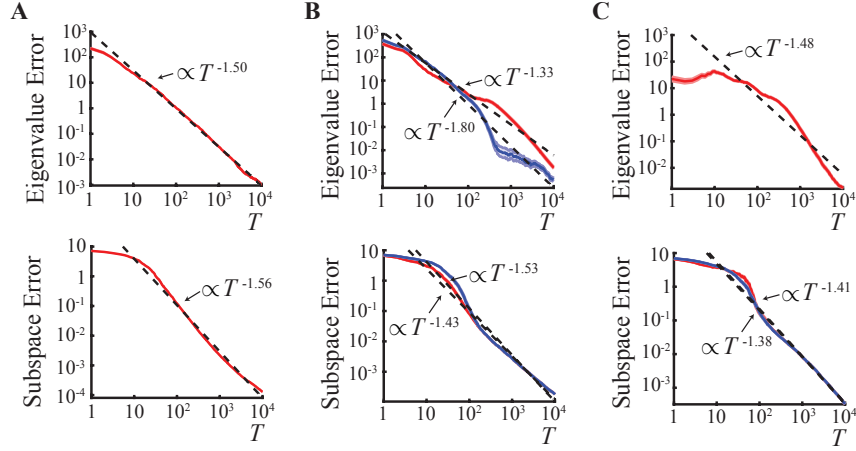

Figure 2: Performance of the three neural networks: soft-thresholding (**A**), hard-thresholding (**B**), equalization after thresholding (**C**). Top: eigenvalue error, bottom: subspace error as a function of data presentations. Solid lines - means and shades - stds over 10 runs. Red - principal, blue - inter-neurons. Dashed lines - best-fit power laws. For metric definitions see text.

time $T$, the algorithm, first, iterates (24) by the dynamics of neuronal activities until convergence and, second, updates synaptic weights according to local anti-Hebbian (for synapses from interneurons) and Hebbian (25) (for all other synapses) rules.

While an algorithm similar to (24), (25), but with predetermined learning rates, was previously given in [15, 14], it has not been derived from an optimization problem. Plumbley's convergence analysis of his algorithm [14] suggests that at the fixed point of synaptic updates, the interneuron activity is also a projection onto the principal subspace. This result is a special case of our offline solution, (9), supported by the online numerical simulations (next Section).

## 4    Numerical simulations

Here, we evaluate the performance of the three online algorithms on a synthetic dataset, which is generated by an $n = 64$ dimensional colored Gaussian process with a specified covariance matrix. In this covariance matrix, the eigenvalues, $\lambda_{1..4} = \{5, 4, 3, 2\}$ and the remaining $\lambda_{5..60}$ are chosen uniformly from the interval $[0, 0.5]$. Correlations are introduced in the covariance matrix by generating random orthonormal eigenvectors. For all three algorithms, we choose $\alpha = 1$ and, for the equalizing algorithm, we choose $\beta = 1$. In all simulated networks, the number of principal neurons, $k = 20$, and, for the hard-thresholding and the equalizing algorithms, the number of interneurons, $l = 5$. Synaptic weight matrices were initialized randomly, and synaptic update learning rates, $1/D_{0,i}^Y$ and $1/D_{0,i}^Z$ were initialized to 0.1. Network dynamics is run with a weight $\eta = 0.1$ until the relative change in $\mathbf{y}_T$ and $\mathbf{z}_T$ in one cycle is $< 10^{-5}$.

To quantify the performance of these algorithms, we use two different metrics. The first metric, eigenvalue error, measures the deviation of output covariance eigenvalues from their optimal offline values given in Theorems 1, 2 and 3. The eigenvalue error at time $T$ is calculated by summing squared differences between the eigenvalues of $\frac{1}{T}\mathbf{Y}\mathbf{Y}^\top$ or $\frac{1}{T}\mathbf{Z}\mathbf{Z}^\top$, and their optimal offline values at time $T$. The second metric, subspace error, quantifies the deviation of the learned subspace from the true principal subspace. To form such metric, at each $T$, we calculate the linear transformation that maps inputs, $\mathbf{x}_T$, to outputs, $\mathbf{y}_T = \mathbf{F}_T^{YX}\mathbf{x}_T$ and $\mathbf{z}_T = \mathbf{F}_T^{ZX}\mathbf{x}_T$, at the fixed points of the neural dynamics stages ((13), (19), (24)) of the three algorithms. Exact expressions for these matrices for all algorithms are given in the Supplementary Material. Then, at each $T$, the deviation is $\left\|\mathbf{F}_{m,T}\mathbf{F}_{m,T}^\top - \mathbf{U}_{m,T}^X\mathbf{U}_{m,T}^{X\top}\right\|_F^2$, where $\mathbf{F}_{m,T}$ is an $n \times m$ matrix whose columns are the top $m$ right singular vectors of $\mathbf{F}_T$, $\mathbf{F}_{m,T}\mathbf{F}_{m,T}^\top$ is the projection matrix to the subspace spanned by these singular vectors, $\mathbf{U}_{m,T}^X$ is an $n \times m$ matrix whose columns are the principal eigenvectors of the input covariance matrix $\mathbf{C}$ at time $T$, $\mathbf{U}_{m,T}^X\mathbf{U}_{m,T}^{X\top}$ is the projection matrix to the principal subspace.

Further numerical simulations comparing the performance of the soft-thresholding algorithm with $\alpha = 0$ with other neural principal subspace algorithms can be found in [24].

## 5  Discussion and conclusions

We developed a normative approach for dimensionality reduction by formulating three novel optimization problems, the solutions of which project the input onto its principal subspace, and rescale the data by i) soft-thresholding, ii) hard-thresholding, iii) equalization after thresholding of the input eigenvalues. Remarkably we found that these optimization problems can be solved online using biologically plausible neural circuits. The dimensionality of neural activity is the number of either input covariance eigenvalues above the threshold, $m$, (if $m < k$) or output neurons, $k$ (if $k \leq m$). The former case is ubiquitous in the analysis of experimental recordings, for a review see [31].

Interestingly, the division of neurons into two populations, principal and interneurons, in the last two models has natural parallels in biological neural networks. In biology, principal neurons and interneurons usually are excitatory and inhibitory respectively. However, we cannot make such an assignment in our theory, because the signs of neural activities, $\mathbf{x}_T$ and $\mathbf{y}_T$, and, hence, the signs of synaptic weights, $\mathbf{W}$, are unconstrained. Previously, interneurons were included into neural circuits [32], [33] outside of the normative approach.

Similarity matching in the offline setting has been used to analyze experimentally recorded neuron activity lending support to our proposal. Semantically similar stimuli result in similar neural activity patterns in human (fMRI) and monkey (electrophysiology) IT cortices [34, 35]. In addition, [36] computed similarities among visual stimuli by matching them with the similarity among corresponding retinal activity patterns (using an information theoretic metric).

We see several possible extensions to the algorithms presented here: 1) Our online objective functions may be optimized by alternative algorithms, such as gradient descent, which map onto different circuit architectures and learning rules. Interestingly, gradient descent-ascent on convex-concave objectives has been previously related to the dynamics of principal and interneurons [37]. 2) Inputs coming from a non-stationary distribution (with time-varying covariance matrix) can be processed by algorithms derived from the objective functions where contributions from older data points are "forgotten", or "discounted". Such discounting results in higher learning rates in the corresponding online algorithms, even at large $T$, giving them the ability to respond to variations in data statistics [24, 4]. Hence, the output dimensionality can track the number of input dimensions whose eigenvalues exceed the threshold. 3) In general, the output of our algorithms is not decorrelated. Such decorrelation can be achieved by including a correlation-penalizing term in our objective functions [38]. 4) Choosing the threshold parameter $\alpha$ requires an a priori knowledge of input statistics. A better solution, to be presented elsewhere, would be to let the network adjust such threshold adaptively, e.g. by filtering out all the eigenmodes with power below the mean eigenmode power. 5) Here, we focused on dimensionality reduction using only spatial, as opposed to the spatio-temporal, correlation structure.

We thank L. Greengard, A. Sengupta, A. Grinshpan, S. Wright, A. Barnett and E. Pnevmatikakis.

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
