[Supplementary Material]

# A Normative Theory of Adaptive Dimensionality Reduction in Neural Networks
## Supplementary Information

Cengiz Pehlevan and Dmitri B. Chklovskii

*Simons Center for Data Analysis, Simons Foundation, New York, NY 10010*

## Contents

## I. OPTIMAL ORTHOGONAL TRANSFORMATIONS FOR DIAGONAL MATRIX ALIGNMENT

Here we reproduce Schur's lemma [1] and prove two new lemmas that will be central to our analysis in the coming sections.

**Lemma** (Schur's lemma). *Let $\lambda_1 \geq \ldots \geq \lambda_p$ and $\mathbf{D}$ a $p \times p$ dimensional doubly stochastic*

*matrix, i.e. a non-negative matrix whose rows and columns separately add to one. Then*

$$\sum_{ij} \lambda_i \hat{\lambda}_j D_{ij} \leq \sum_{j=1}^{p} \hat{\lambda}_j \lambda_j. \tag{S.1}$$

*Proof.*

$$\sum_{ij} \lambda_i \hat{\lambda}_j D_{ij} = \sum_{j=1}^{p} \hat{\lambda}_j \left( \sum_{i=1}^{p} \lambda_i D_{ij} \right)$$

$$= \sum_{j=1}^{p} \left( \hat{\lambda}_j - \hat{\lambda}_{j+1} \right) \sum_{k=1}^{j} \left( \sum_{i=1}^{p} \lambda_i D_{ik} \right) \qquad (\hat{\lambda}_{p+1} := 0)$$

$$= \sum_{j=1}^{p} \left( \hat{\lambda}_j - \hat{\lambda}_{j+1} \right) \sum_{i=1}^{p} \lambda_i \sum_{k=1}^{j} D_{ik}$$

$$= \sum_{j=1}^{p} \left( \hat{\lambda}_j - \hat{\lambda}_{j+1} \right) \left( \sum_{i=1}^{p} (\lambda_i - \lambda_j) \sum_{k=1}^{j} D_{ik} + \lambda_j j \right) \qquad \text{(using } \sum_{i=1}^{p} D_{ik} = 1)$$

$$\leq \sum_{j=1}^{p} \left( \hat{\lambda}_j - \hat{\lambda}_{j+1} \right) \left( \sum_{i=1}^{j} (\lambda_i - \lambda_j) \sum_{k=1}^{j} D_{ik} + \lambda_j j \right) \qquad \text{(using } \lambda_{i>j} - \lambda_j \leq 0)$$

$$= \sum_{j=1}^{p} \left( \hat{\lambda}_j - \hat{\lambda}_{j+1} \right) \left( \sum_{i=1}^{j} \lambda_i \sum_{k=1}^{j} D_{ik} + \lambda_j \sum_{i=1}^{j} \left( 1 - \sum_{k=1}^{j} D_{ik} \right) \right)$$

$$\leq \sum_{j=1}^{p} \left( \hat{\lambda}_j - \hat{\lambda}_{j+1} \right) \left( \sum_{i=1}^{j} \lambda_i \sum_{k=1}^{j} D_{ik} + \sum_{i=1}^{j} \lambda_i \left( 1 - \sum_{k=1}^{j} D_{ik} \right) \right)$$

$$\text{(using } \lambda_{i \leq j} - \lambda_j \geq 0 \text{ and } 1 - \sum_{k=1}^{j} D_{ik} \geq 0)$$

$$= \sum_{j=1}^{p} \left( \hat{\lambda}_j - \hat{\lambda}_{j+1} \right) \sum_{i=1}^{j} \lambda_i$$

$$= \sum_{j=1}^{p} \hat{\lambda}_j \lambda_j. \tag{S.2}$$

$$\square$$

**Lemma 1.** *Let* $\mathbf{\Lambda} = \operatorname{diag}(\lambda_1, \ldots, \lambda_p)$, *where* $\lambda_1 \geq \ldots \geq \lambda_p$ *are real numbers, and let* $\hat{\mathbf{\Lambda}} = \operatorname{diag}\left( \hat{\lambda}_1, \ldots, \hat{\lambda}_p \right)$, *where* $\hat{\lambda}_1 \geq \ldots \geq \hat{\lambda}_p$ *are real numbers. Then,*

$$\max_{\mathbf{O} \in O(p)} \operatorname{Tr}\left( \mathbf{\Lambda} \mathbf{O} \hat{\mathbf{\Lambda}} \mathbf{O}^\top \right) = \operatorname{Tr}\left( \mathbf{\Lambda} \hat{\mathbf{\Lambda}} \right), \tag{S.3}$$

*where* $O(p)$ *is the set of* $p \times p$ *orthogonal matrices.*

*Proof.* To prove the lemma, it is convenient to express the cost in terms of matrix elements:

$$\operatorname{Tr}\left( \mathbf{\Lambda} \mathbf{O} \hat{\mathbf{\Lambda}} \mathbf{O}^\top \right) = \sum_{i,j} \lambda_i \hat{\lambda}_j O_{ij}^2 \tag{S.4}$$

Now consider a matrix $\mathbf{D}$, whose elements are given by $D_{ij} = O_{ij}^2$. Because $\mathbf{O}$ is orthogonal, $\mathbf{D}$ is doubly stochastic: $\sum_i D_{ij} = \sum_i O_{ij}^2 = \left[\mathbf{O}^\top\mathbf{O}\right]_{jj} = 1$ and $\sum_j D_{ij} = \sum_j O_{ij}^2 = \left[\mathbf{O}\mathbf{O}^\top\right]_{ii} = 1$. For any doubly stochastic matrix $\mathbf{D}$, and decreasingly ordered $\{\lambda_i\}$ and $\{\hat{\lambda}_i\}$ according to Schur's lemma:

$$\sum_{i,j} \lambda_i \hat{\lambda}_j D_{ij} \leq \sum_{j=1}^{p} \hat{\lambda}_j \lambda_j. \tag{S.5}$$

Using (S.4) and (S.5), we can conclude that

$$\mathrm{Tr}\left(\boldsymbol{\Lambda}\mathbf{O}\hat{\boldsymbol{\Lambda}}\mathbf{O}^\top\right) \leq \sum_{i=1}^{p} \lambda_i \hat{\lambda}_i. \tag{S.6}$$

The bound is saturated when $\mathbf{O} = \mathbf{I}_p$, which proves the Lemma 1. $\qquad\square$

**Lemma 2.** *Let* $\boldsymbol{\Lambda} = \mathrm{diag}\left(\lambda_1, \ldots, \lambda_p\right)$, *where* $\lambda_1 \geq \ldots \geq \lambda_p$ *are real numbers, and let* $\hat{\boldsymbol{\Lambda}} = \mathrm{diag}\left(\hat{\lambda}_1, \ldots, \hat{\lambda}_p\right)$, *where* $\hat{\lambda}_1 \geq \ldots \geq \hat{\lambda}_p$ *are real numbers. Then,*

$$\underset{\mathbf{O}\in O(p)}{\arg\min} \left\|\boldsymbol{\Lambda} - \mathbf{O}\hat{\boldsymbol{\Lambda}}\mathbf{O}^\top\right\|_F^2 = \underset{\mathbf{O}\in O(p)}{\arg\max} \mathrm{Tr}\left(\boldsymbol{\Lambda}\mathbf{O}\hat{\boldsymbol{\Lambda}}\mathbf{O}^\top\right), \tag{S.7}$$

*where* $O(p)$ *is the set of* $p \times p$ *orthogonal matrices. Furthermore, an orthogonal matrix is optimal if and only if it can be written as a product of two orthogonal matrices*

$$\mathbf{O}^* = \mathbf{O}_{\boldsymbol{\Lambda}}\mathbf{O}_{\hat{\boldsymbol{\Lambda}}}, \tag{S.8}$$

*which commute with* $\boldsymbol{\Lambda}$ *and* $\hat{\boldsymbol{\Lambda}}$ *respectively:*

$$[\boldsymbol{\Lambda}, \mathbf{O}_{\boldsymbol{\Lambda}}] = 0, \qquad \left[\hat{\boldsymbol{\Lambda}}, \mathbf{O}_{\hat{\boldsymbol{\Lambda}}}\right] = 0. \tag{S.9}$$

*Proof.* The first equality in (S.7) follows from the definition of the Frobenius norm and orthogonality of $\mathbf{O}$:

$$\begin{aligned}
\underset{\mathbf{O}\in O(p)}{\arg\min} \left\|\boldsymbol{\Lambda} - \mathbf{O}\hat{\boldsymbol{\Lambda}}\mathbf{O}^\top\right\|_F^2 &= \underset{\mathbf{O}\in O(p)}{\arg\min} \mathrm{Tr}\left[\left(\boldsymbol{\Lambda} - \mathbf{O}\hat{\boldsymbol{\Lambda}}\mathbf{O}^\top\right)^\top \left(\boldsymbol{\Lambda} - \mathbf{O}\hat{\boldsymbol{\Lambda}}\mathbf{O}^\top\right)\right] \\
&= \underset{\mathbf{O}\in O(p)}{\arg\min} \mathrm{Tr}\left[\boldsymbol{\Lambda}^2 - 2\mathbf{O}\hat{\boldsymbol{\Lambda}}\mathbf{O}^\top\boldsymbol{\Lambda} + \hat{\boldsymbol{\Lambda}}^2\right] \\
&= \underset{\mathbf{O}\in O(p)}{\arg\max} \mathrm{Tr}\left(\boldsymbol{\Lambda}\mathbf{O}\hat{\boldsymbol{\Lambda}}\mathbf{O}^\top\right). \tag{S.10}
\end{aligned}$$

It is easy to see that any matrix of the form (S.8) optimizes (S.10):

$$\mathrm{Tr}\left(\boldsymbol{\Lambda}\mathbf{O}_{\boldsymbol{\Lambda}}\mathbf{O}_{\hat{\boldsymbol{\Lambda}}}\hat{\boldsymbol{\Lambda}}\mathbf{O}_{\hat{\boldsymbol{\Lambda}}}^\top\mathbf{O}_{\boldsymbol{\Lambda}}^\top\right) = \mathrm{Tr}\left(\mathbf{O}_{\boldsymbol{\Lambda}}\boldsymbol{\Lambda}\hat{\boldsymbol{\Lambda}}\mathbf{O}_{\hat{\boldsymbol{\Lambda}}}\mathbf{O}_{\hat{\boldsymbol{\Lambda}}}^\top\mathbf{O}_{\boldsymbol{\Lambda}}^\top\right) = \mathrm{Tr}\left(\boldsymbol{\Lambda}\hat{\boldsymbol{\Lambda}}^\top\right), \tag{S.11}$$

which optimizes (S.10) according to Lemma 1.

To prove that all optimal orthogonal matrices are of the form (S.8), we take the following steps:

1. Recall that any orthogonal matrix $\mathbf{O}$ must have $\det \mathbf{O} = \pm 1$. Orthogonal matrices with $\det \mathbf{O} = 1$ are proper rotations which we denote by $\mathbf{R}$. Orthogonal matrices with $\det \mathbf{O} = -1$ are improper rotations which we denote by $\bar{\mathbf{R}}$.

   Without loss of generality it suffices to prove our claim for proper rotations only. As we show now, if all optimal proper rotations are of the form (S.8), then all optimal improper rotations are also of the form (S.8).

   Consider an optimal improper rotation $\bar{\mathbf{R}}^*$:

   $$\mathrm{Tr}\left( \boldsymbol{\Lambda} \bar{\mathbf{R}}^* \hat{\boldsymbol{\Lambda}} \bar{\mathbf{R}}^{*\top} \right) = \mathrm{Tr}\left( \boldsymbol{\Lambda} \hat{\boldsymbol{\Lambda}} \right). \tag{S.12}$$

   Next, we define a one-to-one mapping between each improper rotation $\bar{\mathbf{R}}$ and a proper rotation by multiplying $\bar{\mathbf{R}}$ on the right by the matrix $\mathrm{diag}\left( -1, 1, \ldots, 1 \right)$

   $$\mathbf{R} \equiv \mathrm{diag}\left( -1, 1, \ldots, 1 \right) \bar{\mathbf{R}}. \tag{S.13}$$

   Then,

   $$\mathrm{Tr}\left( \boldsymbol{\Lambda} \mathbf{R}^* \hat{\boldsymbol{\Lambda}} \mathbf{R}^{*\top} \right) = \mathrm{Tr}\left( \boldsymbol{\Lambda} \bar{\mathbf{R}}^* \hat{\boldsymbol{\Lambda}} \bar{\mathbf{R}}^{*\top} \right) = \mathrm{Tr}\left( \boldsymbol{\Lambda} \hat{\boldsymbol{\Lambda}} \right), \tag{S.14}$$

   and therefore $\mathbf{R}^*$ is also optimal. If $\mathbf{R}^*$ is of the form (S.8)

   $$\mathbf{R}^* = \mathbf{O}_{\boldsymbol{\Lambda}} \mathbf{O}_{\hat{\boldsymbol{\Lambda}}}, \tag{S.15}$$

   then corresponding $\bar{\mathbf{R}}^*$,

   $$\bar{\mathbf{R}}^* = \mathrm{diag}\left( -1, 1, \ldots, 1 \right) \mathbf{O}_{\boldsymbol{\Lambda}} \mathbf{O}_{\hat{\boldsymbol{\Lambda}}} \tag{S.16}$$

   is also of the form (S.8), since $\mathrm{diag}\left( -1, 1, \ldots, 1 \right) \mathbf{O}_{\boldsymbol{\Lambda}}$ commutes with $\boldsymbol{\Lambda}$.

   Hence, we only consider proper rotation matrices without loss of generality.

2. Consider an optimal proper rotation matrix $\mathbf{R}^*$:

   $$\mathrm{Tr}\left( \boldsymbol{\Lambda} \mathbf{R}^* \hat{\boldsymbol{\Lambda}} \mathbf{R}^{*\top} \right) = \mathrm{Tr}\left( \boldsymbol{\Lambda} \hat{\boldsymbol{\Lambda}} \right). \tag{S.17}$$

Proper rotations form a connected set, and can be parametrized by $e^{\mathbf{A}}$, where $\mathbf{A}$ is an antisymmetric matrix. Suppose we rotate $\mathbf{R}^*$ by an infinitesimal amount, i.e. $e^{\delta \mathbf{A}} \mathbf{R}^*$:

$$
\begin{aligned}
\mathrm{Tr}\left(\mathbf{\Lambda} e^{\delta\mathbf{A}} \mathbf{R}^* \hat{\mathbf{\Lambda}} \mathbf{R}^{*\top} e^{-\delta\mathbf{A}}\right) &= \mathrm{Tr}\left(\mathbf{\Lambda}\left(\mathbf{I}+\delta\mathbf{A}\right)\mathbf{R}^*\hat{\mathbf{\Lambda}}\mathbf{R}^{*\top}\left(\mathbf{I}-\delta\mathbf{A}\right)\right) + \mathcal{O}(\delta^2) \\
&= \mathrm{Tr}\left(\mathbf{\Lambda}\mathbf{R}^*\hat{\mathbf{\Lambda}}\mathbf{R}^{*\top}\right) \\
&\quad + \mathrm{Tr}\left(\delta\mathbf{A}\left(\mathbf{R}^*\hat{\mathbf{\Lambda}}\mathbf{R}^{*\top}\mathbf{\Lambda} - \mathbf{\Lambda}\mathbf{R}^*\hat{\mathbf{\Lambda}}\mathbf{R}^{*\top}\right)\right) + \mathcal{O}(\delta^2) \quad \text{(S.18)}
\end{aligned}
$$

Since $\mathbf{R}^*$ is maximal, the change in left had side must vanish to first order in $\delta\mathbf{A}$:

$$
\begin{aligned}
0 &= \mathrm{Tr}\left(\delta\mathbf{A}\left(\mathbf{R}^*\hat{\mathbf{\Lambda}}\mathbf{R}^{*\top}\mathbf{\Lambda} - \mathbf{\Lambda}\mathbf{R}^*\hat{\mathbf{\Lambda}}\mathbf{R}^{*\top}\right)\right) = \sum_{ij}\delta A_{ij}\left[\mathbf{R}^*\hat{\mathbf{\Lambda}}\mathbf{R}^{*\top}\mathbf{\Lambda} - \mathbf{\Lambda}\mathbf{R}^*\hat{\mathbf{\Lambda}}\mathbf{R}^{*\top}\right]_{ji} \\
&= 2\sum_{i,j<i}\delta A_{ij}\left[\mathbf{R}^*\hat{\mathbf{\Lambda}}\mathbf{R}^{*\top}\mathbf{\Lambda} - \mathbf{\Lambda}\mathbf{R}^*\hat{\mathbf{\Lambda}}\mathbf{R}^{*\top}\right]_{ji}, \quad\quad\quad\quad\quad\quad \text{(S.19)}
\end{aligned}
$$

where we used the antisymmetry of $\delta\mathbf{A}$ and of $\left(\mathbf{R}^*\hat{\mathbf{\Lambda}}\mathbf{R}^{*\top}\mathbf{\Lambda} - \mathbf{\Lambda}\mathbf{R}^*\hat{\mathbf{\Lambda}}\mathbf{R}^{*\top}\right)$. Since $\delta A_{i,j<i}$ are independent perturbations, their coefficients $\left(\mathbf{R}^*\hat{\mathbf{\Lambda}}\mathbf{R}^{*\top}\mathbf{\Lambda} - \mathbf{\Lambda}\mathbf{R}^*\hat{\mathbf{\Lambda}}\mathbf{R}^{*\top}\right)_{ji}$ must each be zero. From here we conclude that for maximal $\mathbf{R}^*$

$$
\mathbf{\Lambda}\mathbf{R}^*\hat{\mathbf{\Lambda}}\mathbf{R}^{*\top} - \mathbf{R}^*\hat{\mathbf{\Lambda}}\mathbf{R}^{*\top}\mathbf{\Lambda} = \left[\mathbf{\Lambda}, \mathbf{R}^*\hat{\mathbf{\Lambda}}\mathbf{R}^{*\top}\right] = 0. \quad\quad\quad \text{(S.20)}
$$

3. We will use (S.20) to prove our claim that all optimal proper rotation matrices are of the form (S.8). We remind that if a matrix commutes with a diagonal matrix, it must be block-diagonal. There is a separate block for each distinct diagonal element of the diagonal matrix, and the size of the block is given by the degeneracy of the diagonal element. Then,

$$
\mathbf{B_\Lambda} \equiv \mathbf{R}^*\hat{\mathbf{\Lambda}}\mathbf{R}^{*\top} \quad\quad\quad\quad\quad\quad\quad\quad \text{(S.21)}
$$

is block diagonal with with blocks defined by the degenerate diagonal elements of $\mathbf{\Lambda}$. Further, singular values of $\mathbf{B_\Lambda}$ are given by diagonals of $\hat{\mathbf{\Lambda}}$.

$\mathbf{B_\Lambda}$ can be diagonalized by another orthogonal matrix., which is block diagonal with the same blocks as in $\mathbf{B_\Lambda}$. However, such a block diagonal matrix would also commute with $\mathbf{\Lambda}$. Then, with notation from (S.8),

$$
\mathbf{B_\Lambda} = \mathbf{O_\Lambda}\hat{\mathbf{\Lambda}}\mathbf{O_\Lambda^\top}. \quad\quad\quad\quad\quad\quad\quad\quad \text{(S.22)}
$$

(S.21) and (S.22) imply

$$\left[\hat{\mathbf{\Lambda}}, \mathbf{O}_{\mathbf{\Lambda}}^{\top}\mathbf{R}^*\right] = 0. \tag{S.23}$$

$\mathbf{O}_{\mathbf{\Lambda}}^{\top}\mathbf{R}^*$ is block diagonal with blocks defined by the degenerate diagonal elements of $\hat{\mathbf{\Lambda}}$. With notation from (S.8),

$$\mathbf{O}_{\mathbf{\Lambda}}^{\top}\mathbf{R}^* = \mathbf{O}_{\hat{\mathbf{\Lambda}}}, \tag{S.24}$$

and hence

$$\mathbf{R}^* = \mathbf{O}_{\mathbf{\Lambda}}\mathbf{O}_{\hat{\mathbf{\Lambda}}}. \tag{S.25}$$

Therefore we can conclude that all rotation matrices that optimize (S.3) are of the form (S.8).

$\square$

## II.   PROOF OF THEOREM 1 - SOFT-THRESHOLDING OF COVARIANCE EIGENVALUES

We reproduce the offline objective function (3) for ease of referencing.

$$\min_{\mathbf{Y}} \left\|\mathbf{X}^{\top}\mathbf{X} - \mathbf{Y}^{\top}\mathbf{Y} - \alpha T\mathbf{I}_T\right\|_F^2, \tag{S.26}$$

where $\alpha \geq 0$, $\mathbf{X} \in \mathbb{R}^{n \times T}$ and $\mathbf{Y} \in \mathbb{R}^{k \times T}$. Define $m$ to be the number of eigenvalues of $\mathbf{C} = \frac{1}{T}\mathbf{X}\mathbf{X}^{\top}$ greater than or equal to $\alpha$

Now, we present the main result of this subsection and its proof.

**Theorem 1.** *Suppose an eigen-decomposition of $\mathbf{X}^{\top}\mathbf{X} = \mathbf{V}^X\mathbf{\Lambda}^X\mathbf{V}^{X^{\top}}$, where $\mathbf{\Lambda}^X = \mathrm{diag}\left(\lambda_1^X, \ldots, \lambda_T^X\right)$ with $\lambda_1^X \geq \ldots \geq \lambda_T^X$. Note that $\mathbf{\Lambda}^X$ has at most $n$ nonzero eigenvalues coinciding with those of $T\mathbf{C}$. Then,*

$$\mathbf{Y}^* = \mathbf{U}_k \, \mathbf{ST}_k(\mathbf{\Lambda}^X, \alpha T)^{1/2} \, \mathbf{V}_k^{X^{\top}}, \tag{S.27}$$

*are optima of (S.26), where $\mathbf{ST}_k(\mathbf{\Lambda}^X, \alpha T) = \mathrm{diag}\left(\mathrm{ST}\left(\lambda_1^X, \alpha T\right), \ldots, \mathrm{ST}\left(\lambda_k^X, \alpha T\right)\right)$, ST is the soft-thresholding function, $\mathrm{ST}(a, b) = \max(a - b, 0)$, $\mathbf{V}_k^X$ consists of the columns of $\mathbf{V}^X$ corresponding to the top $k$ eigenvalues, i.e. $\mathbf{V}_k^X = \left[\mathbf{v}_1^X, \ldots, \mathbf{v}_k^X\right]$ and $\mathbf{U}_k$ is any $k \times k$ orthogonal matrix, i.e. $\mathbf{U}_k \in O(k)$. The form (S.27) uniquely defines all optima of (S.26), except when $k < m$, $\lambda_k^X > \alpha T$ and $\lambda_k^X = \lambda_{k+1}^X$.*

*Proof.* Here we assume that if $k < m$ and $\lambda_k^X > \alpha T$, then $\lambda_k^X \neq \lambda_{k+1}^X$, and prove that the form (S.27) uniquely defines all optima of (S.26). The exceptional case of $k < m$, $\lambda_k^X > \alpha T$ and $\lambda_k^X = \lambda_{k+1}^X$ is treated in a remark below.

Since the cost (S.26) depends on $\mathbf{Y}$ only through the similarity matrix $\mathbf{Y}^\top\mathbf{Y}$, we first optimize (S.26) with respect to $\mathbf{Y}^\top\mathbf{Y}$ and then reconstruct the optimal $\mathbf{Y}$. In turn we optimize with respect to $\mathbf{Y}^\top\mathbf{Y}$ considering eigendecomposition of $\mathbf{Y}^\top\mathbf{Y} = \mathbf{V}^Y\mathbf{\Lambda}^Y\mathbf{V}^{Y^\top}$, and finding optimal $\mathbf{V}^Y$ and $\mathbf{\Lambda}^Y$ separately.

We first optimize (S.26) with respect to $\mathbf{V}^Y \in O(T)$ for fixed $\mathbf{\Lambda}^Y$. Because the Frobenius norm is invariant to orthogonal rotations, for any $\mathbf{Y}^\top\mathbf{Y}$, the objective (S.26) can be rewritten as

$$\left\|\mathbf{X}^\top\mathbf{X} - \mathbf{Y}^\top\mathbf{Y} - \alpha T\mathbf{I}_T\right\|_F^2 = \left\|\mathbf{\Lambda}^X - \mathbf{O}\mathbf{\Lambda}^Y\mathbf{O}^\top - \alpha T\mathbf{I}_T\right\|_F^2, \tag{S.28}$$

where $\mathbf{O} = \mathbf{V}^{X^\top}\mathbf{V}^Y \in O(T)$. The minimization with respect to $\mathbf{V}^Y$ is equivalent to a minimization over $\mathbf{O}$ from which $\mathbf{V}^Y$ can be recovered uniquely by $\mathbf{V}^X\mathbf{O}$. According to Lemma 2, each orthogonal matrix $\mathbf{O} = \mathbf{V}^{X^\top}\mathbf{V}^Y$ that is a product of two orthogonal matrices $\mathbf{O} = \mathbf{O}_{\mathbf{\Lambda}^X}\mathbf{O}_{\mathbf{\Lambda}^Y}$ with $\left[\mathbf{\Lambda}^X - \alpha T\mathbf{I}_T, \mathbf{O}_{\mathbf{\Lambda}^X}\right] = 0$ and $\left[\mathbf{\Lambda}^Y, \mathbf{O}_{\mathbf{\Lambda}^Y}\right] = 0$, is optimal. Then, optimal $\mathbf{V}^Y$ is given by

$$\mathbf{V}^{Y*} = \mathbf{V}^X\mathbf{O}_{\mathbf{\Lambda}^X}\mathbf{O}_{\mathbf{\Lambda}^Y}. \tag{S.29}$$

and the optimal value of (S.28) is:

$$\sum_{i=1}^{T} \left(\lambda_i^X - \lambda_i^Y - \alpha T\right)^2. \tag{S.30}$$

It remains to find optimal $\mathbf{\Lambda}^Y$, which minimizes (S.30):

$$\min_{\lambda_1^Y,\dots,\lambda_T^Y} \sum_{i=1}^{T} \left(\lambda_i^X - \lambda_i^Y - \alpha T\right)^2, \tag{S.31}$$

where $\{\lambda_1^Y,\dots,\lambda_T^Y\}$ are non-negative and at most $k$ of them are non-zero. Consider a term $\left(\lambda_i^X - \lambda_i^Y - \alpha T\right)^2$ in the sum. If $\lambda_i^X \leq \alpha T$, choosing a positive $\lambda_i^Y$ will only increase the term, hence optimal $\lambda_i^Y = 0$ for such terms. If $\lambda_i^X > \alpha T$, then choosing $\lambda_i^Y = \lambda_i^X - \alpha T$ will set the term to 0, i.e. its minimum. On the other hand, at most $k$ of $\{\lambda_1^Y,\dots,\lambda_T^Y\}$ can be non-zero. These $k$ eigenvalues should be allocated to largest non-negative values of $\lambda_i^X - \alpha T$.

Therefore, optimal $\{\lambda_1^Y, \ldots, \lambda_T^Y\}$ are

$$
\lambda_i^{Y*} = \begin{cases} \text{ST}\left(\lambda_i^X, \alpha T\right), & i \leq k \\ 0, & i > k \end{cases}. \tag{S.32}
$$

To reconstruct $\mathbf{Y}^*$, using (S.29) and (S.32) we rewrite the eigenvalue decomposition of $\mathbf{Y}^{*\top}\mathbf{Y}^*$. Using $\mathbf{\Lambda}^{Y*}$ to denote optimal singular values defined by (S.32), and $\mathbf{O}_{\mathbf{\Lambda}^{Y*}}$ to denote an orthogonal matrix that commutes with $\mathbf{\Lambda}^{Y*}$, we get:

$$
\begin{aligned}
\mathbf{Y}^{*\top}\mathbf{Y}^* &= \mathbf{V}^X \mathbf{O}_{\mathbf{\Lambda}^X} \mathbf{O}_{\mathbf{\Lambda}^{Y*}} \mathbf{\Lambda}^{Y*} \mathbf{O}_{\mathbf{\Lambda}^{Y*}}^\top \mathbf{O}_{\mathbf{\Lambda}^X}^\top \mathbf{V}^{X\top} \\
&= \mathbf{V}^X \mathbf{O}_{\mathbf{\Lambda}^X} \mathbf{\Lambda}^{Y*} \mathbf{O}_{\mathbf{\Lambda}^X}^\top \mathbf{V}^{X\top}.
\end{aligned} \tag{S.33}
$$

Since if diagonal elements of $\mathbf{\Lambda}^X$ are degenerate, the corresponding diagonal elements of $\mathbf{\Lambda}^{Y*}$ must be degenerate

$$
\left[\mathbf{\Lambda}^{Y*}, \mathbf{O}_{\mathbf{\Lambda}^X}\right] = 0. \tag{S.34}
$$

Hence,

$$
\mathbf{Y}^{*\top}\mathbf{Y}^* = \mathbf{V}^X \mathbf{\Lambda}^{Y*} \mathbf{V}^{X\top}. \tag{S.35}
$$

These $\mathbf{Y}$ matrices can be constructed as in (S.27): its columns are coordinates in the arbitrarily rotated orthogonal basis spanning the $k$-dimensional principal subspace of $\mathbf{X}\mathbf{X}^\top$.

$\square$

*Remark* 1. In the case $k < m$, $\lambda_k^X > \alpha T$ and $\lambda_k^X = \lambda_{k+1}^X$, (S.34) is not generally true anymore, because while $\lambda_k^X = \lambda_{k+1}^X$, $\lambda_k^Y \neq \lambda_{k+1}^Y$. $\mathbf{Y}$ matrices constructed as in (S.27) are still minima, as can be seen by choosing $\mathbf{O}_{\mathbf{\Lambda}^X} = \mathbf{I}_T$ for which (S.34) holds, but there are other solutions which cannot be put in the form (S.27). We observe that blocks of $\mathbf{O}_{\mathbf{\Lambda}^X}$ that do not correspond to $\lambda_k^X$ still commute with $\mathbf{\Lambda}^{Y*}$. Thus, when $k < m$, $\lambda_k^X > \alpha T$ and $\lambda_k^X = \lambda_{k+1}^X$, we can write the most general solution as

$$
\mathbf{Y}^* = \mathbf{U}_k \, \mathbf{\Lambda}_{k \times T}^{Y* \, 1/2} \, \mathbf{O}_{\mathbf{\Lambda}_k^X}^\top \mathbf{V}^{X\top}, \tag{S.36}
$$

where $\mathbf{U}_k$ is a $k \times k$ orthogonal matrix, $\mathbf{\Lambda}_{k \times T}^{Y*}$ is a $k \times T$ diagonal matrix with its $k$ diagonals set to first $k$ diagonals of $\mathbf{\Lambda}^{Y*}$, and $\mathbf{O}_{\mathbf{\Lambda}_k^X}$ is a $T \times T$ orthogonal matrix that is diagonal except one block that corresponds to diagonal elements of $\mathbf{\Lambda}^X$ that are degenerate with $\lambda_k^X$.

### III. PROOF OF THEOREM 2 - HARD-THRESHOLDING OF COVARIANCE EIGENVALUES

We reproduce the offline objective (6) for ease of referencing.

$$\min_{\mathbf{Y}} \max_{\mathbf{Z}} \left\| \mathbf{X}^\top \mathbf{X} - \mathbf{Y}^\top \mathbf{Y} \right\|_F^2 - \left\| \mathbf{Y}^\top \mathbf{Y} - \mathbf{Z}^\top \mathbf{Z} - \alpha T \mathbf{I}_T \right\|_F^2, \tag{S.37}$$

where $\alpha \geq 0$, $\mathbf{X} \in \mathbb{R}^{n \times T}$, $\mathbf{Y} \in \mathbb{R}^{k \times T}$ and $\mathbf{Z} \in \mathbb{R}^{l \times T}$. Let $m$ be the number of eigenvalues $\mathbf{C}$ greater than or equal to $\alpha$.

**Theorem 2.** *Suppose an eigen-decomposition of $\mathbf{X}^\top \mathbf{X} = \mathbf{V}^X \mathbf{\Lambda}^X \mathbf{V}^{X\top}$, where $\mathbf{\Lambda}^X = \mathrm{diag}\left(\lambda_1^X, \ldots, \lambda_T^X\right)$ with $\lambda_1^X \geq \ldots \geq \lambda_T^X \geq 0$. Assume $l \geq \min(k, m)$. Then,*

$$\mathbf{Y}^* = \mathbf{U}_k \, \mathbf{HT}_k(\mathbf{\Lambda}^X, \alpha T)^{1/2} \, \mathbf{V}_k^{X\top}, \qquad \mathbf{Z}^* = \mathbf{U}_l \, \mathbf{ST}_{l,\min(k,m)}(\mathbf{\Lambda}^X, \alpha T)^{1/2} \, \mathbf{V}_l^{X\top}, \tag{S.38}$$

*are optima of (S.37), where $\mathbf{HT}_k(\mathbf{\Lambda}^X, \alpha T) = \mathrm{diag}\left(\mathrm{HT}\left(\lambda_1^X, \alpha T\right), \ldots, \mathrm{HT}\left(\lambda_k^X, \alpha T\right)\right)$, $\mathrm{HT}(a, b) = a\Theta(a - b)$ with $\Theta()$ being the step function: $\Theta(a - b) = 1$ if $a \geq b$ and $\Theta(a - b) = 0$ if $a < b$, $\mathbf{ST}_{l,\min(k,m)}(\mathbf{\Lambda}^X, \alpha T) = \mathrm{diag}\big(\mathrm{ST}\left(\lambda_1^X, \alpha T\right), \ldots, \mathrm{ST}\left(\lambda_{\min(k,m)}^X, \alpha T\right), \underbrace{0, \ldots, 0}_{l - \min(k,m)}\big)$, $\mathbf{V}_p^X = \left[\mathbf{v}_1^X, \ldots, \mathbf{v}_p^X\right]$ and $\mathbf{U}_p \in O(p)$. The form (S.38) uniquely defines all optima (S.37) except when either 1) $\alpha$ is an eigenvalue of $\mathbf{C}$ or 2) $k < m$ and $\lambda_k^X = \lambda_{k+1}^X$.*

*Proof.* Here we assume 1) $l \geq \min(k, m)$, 2) $\alpha$ is not an eigenvalue of $\mathbf{C}$ and 3) if $k < m$, then $\lambda_k^X \neq \lambda_{k+1}^X$. We prove that with these assumptions, the form (S.38) uniquely defines all optima (S.37). Violations of these assumptions are treated in three remarks below.

The proof is similar to that of Theorem 1. Since the objective (S.37) depends on $\mathbf{Y}$ only through the similarity matrix $\mathbf{Y}^\top \mathbf{Y}$ and on $\mathbf{Z}$ through $\mathbf{Z}^\top \mathbf{Z}$, we first find optimal $\mathbf{Y}^\top \mathbf{Y}$ and $\mathbf{Z}^\top \mathbf{Z}$ from which we reconstruct $\mathbf{Y}$ and $\mathbf{Z}$. Our strategy is to start with eigendecompositions of $\mathbf{Y}^\top \mathbf{Y} = \mathbf{V}^Y \mathbf{\Lambda}^Y \mathbf{V}^{Y\top}$ and $\mathbf{Z}^\top \mathbf{Z} = \mathbf{V}^Z \mathbf{\Lambda}^Z \mathbf{V}^{Z\top}$, and find optimal $\mathbf{V}^Y$, $\mathbf{\Lambda}^Y$, $\mathbf{V}^Z$ and $\mathbf{\Lambda}^Z$.

We first optimize (S.37) with respect to $\mathbf{V}^Y \in \mathbb{R}^{T \times T}$ and $\mathbf{V}^Z \in \mathbb{R}^{T \times T}$ for fixed $\mathbf{\Lambda}^Y$ and $\mathbf{\Lambda}^Z$. Because Frobenius norm is invariant under rotations, the terms in (S.37) can be rewritten as

$$\left\| \mathbf{X}^\top \mathbf{X} - \mathbf{Y}^\top \mathbf{Y} \right\|_F^2 = \left\| \mathbf{\Lambda}^X - \mathbf{O} \mathbf{\Lambda}^Y \mathbf{O}^\top \right\|_F^2, \tag{S.39}$$

and

$$-\left\| \mathbf{Y}^\top \mathbf{Y} - \mathbf{Z}^\top \mathbf{Z} - \alpha T \mathbf{I}_T \right\|_F^2 = -\left\| \mathbf{\Lambda}^Y - \mathbf{Q} \mathbf{\Lambda}^Z \mathbf{Q}^\top - \alpha T \mathbf{I}_T \right\|_F^2, \tag{S.40}$$

where $\mathbf{O} = \mathbf{V}^{X\top}\mathbf{V}^Y \in O(T)$ and where $\mathbf{Q} = \mathbf{V}^{Y\top}\mathbf{V}^Z \in O(T)$. First, we maximize (S.37) with respect to $\mathbf{V}^Z$, which enters via $\mathbf{Q}$ in (S.40). According to Lemma 2, any orthogonal matrix $\mathbf{Q} = \mathbf{V}^{Y\top}\mathbf{V}^Z$ that is a product of two orthogonal matrices $\mathbf{Q} = \mathbf{Q}_{\mathbf{\Lambda}^Y}\mathbf{Q}_{\mathbf{\Lambda}^z}$ with $\left[\mathbf{\Lambda}^Y - \alpha T\mathbf{I}_T, \mathbf{Q}_{\mathbf{\Lambda}^Y}\right] = 0$ and $\left[\mathbf{\Lambda}^Z, \mathbf{Q}_{\mathbf{\Lambda}^z}\right] = 0$, is optimal. Then, optimal $\mathbf{V}^Z$ is given by

$$\mathbf{V}^{Z*} = \mathbf{V}^Y\mathbf{Q}_{\mathbf{\Lambda}^Y}\mathbf{Q}_{\mathbf{\Lambda}^z}. \tag{S.41}$$

and subsituting this expression into (S.40),

$$-\sum_{i=1}^{T}\left(\lambda_i^Y - \lambda_i^Z - \alpha T\right)^2. \tag{S.42}$$

Because the optimality of $\mathbf{V}^{Z*}$ (S.41) holds for any $\mathbf{V}^Y$ the minimization of (S.41) with respect to $\mathbf{V}^Y$ is reduced to (S.39). According to Lemma 2, any orthogonal matrix $\mathbf{O} = \mathbf{V}^{X\top}\mathbf{V}^Y$ that is a product of two orthogonal matrices $\mathbf{O} = \mathbf{O}_{\mathbf{\Lambda}^x}\mathbf{O}_{\mathbf{\Lambda}^Y}$ with $\left[\mathbf{\Lambda}^X, \mathbf{O}_{\mathbf{\Lambda}^x}\right] = 0$ and $\left[\mathbf{\Lambda}^Y, \mathbf{O}_{\mathbf{\Lambda}^Y}\right] = 0$, is optimal. Then, optimal $\mathbf{V}^Y$ is given by

$$\mathbf{V}^{Y*} = \mathbf{V}^X\mathbf{O}_{\mathbf{\Lambda}^x}\mathbf{O}_{\mathbf{\Lambda}^Y}. \tag{S.43}$$

For these choices of $\mathbf{V}^{Y*}$ and $\mathbf{V}^{Z*}$, the full objective (S.37) reduces to:

$$\min_{\lambda_1^Y,\ldots,\lambda_T^Y} \max_{\lambda_1^Z,\ldots,\lambda_T^Z} \sum_{i=1}^{T}\left[\left(\lambda_i^X - \lambda_i^Y\right)^2 - \left(\lambda_i^Y - \alpha T - \lambda_i^Z\right)^2\right], \tag{S.44}$$

where $\{\lambda_1^Z,\ldots,\lambda_T^Z\}$ are constrained to be non-negative and at most $l$ of them are non-zero, and $\{\lambda_1^Y,\ldots,\lambda_T^Y\}$ are also constrained to be non-negative and at most $k$ of them are non-zero. We analyze the terms in the sum separately:

1. Consider the $i > m$ terms in the sum for which $\lambda_i^X < \alpha T$. For such terms, choosing $\lambda_i^Y = \lambda_i^Z = 0$ gives the optimal cost, $\lambda_i^{X^2} - \alpha^2 T^2 < 0$. To see this, let's calculate costs associated with other choices of $\lambda_i^Y$ and $\lambda_i^Z$. Suppose $\lambda_i^Y \geq \alpha T$. Then, maximization with respect to $\lambda_i^Z$ would set $\left(\lambda_i^Y - \alpha T - \lambda_i^Z\right)^2 = 0$ and therefore the cost would be $\left(\lambda_i^X - \lambda_i^Y\right)^2 \geq 0$. Suppose $\lambda_i^Y \leq \alpha T$. Then, maximization with respect to $\lambda_i^Z$ would set $\lambda_i^Z = 0$, and the cost would be $\lambda_i^{X^2} - \alpha^2 T^2 + 2\lambda_i^Y\left(\alpha T - \lambda_i^X\right)$. This is minimized for $\lambda_i^Y = 0$. Hence, our claim holds.

2. Consider the $i \leq m$ terms in the sum for which $\lambda_i^X > \alpha T$. Since $\lambda_{i>m}^Y = \lambda_{i>m}^Z = 0$, and we assumed $l \geq m$, we can assign all $\lambda_{i\leq m}^Z$ to non-zero values if needed. On the other hand, $k$ can be less than $m$ and we might be forced to set some $\lambda_{i\leq m}^Y$ to zero.

(a) Suppose $\lambda_i^Y > 0$. For these terms, choosing $\lambda_i^Y = \lambda_i^X$ and $\lambda_i^Z = \lambda_i^X - \alpha T$ gives the optimal cost, 0. To see this, let's calculate costs associated with other choices of $\lambda_i^Y$ and $\lambda_i^Z$. If $\lambda_i^Y > \lambda_i^X$, or $\alpha T \leq \lambda_i^Y < \lambda_i^X$ maximization with respect to $\lambda_i^Z$ would set $\left(\lambda_i^Y - \alpha T - \lambda_i^Z\right)^2 = 0$ and therefore the cost would be $\left(\lambda_i^X - \lambda_i^Y\right)^2 > 0$. If $\lambda_i^Y < \alpha T$, maximization with respect to $\lambda_i^Z$ would set $\lambda_i^Z = 0$, and the cost would be $\lambda_i^{X\,2} - \alpha^2 T^2 - 2\lambda_i^Y\left(\lambda_i^X - \alpha T\right)$. This cost is greater than its value at $\lambda_i^Y = \alpha T$, which is $\left(\lambda_i^X - \alpha T\right)^2 > 0$. Hence, our claim holds.

(b) Suppose $\lambda_i^Y = 0$. For these terms, choosing $\lambda_i^Z = 0$ gives the optimal cost, $\lambda_i^{X\,2} - \alpha^2 T^2 > 0$.

Therefore, one assigns non-zero $\lambda_i^Y$ to the the first $\min(k, m)$ terms in the sum. For such terms $\lambda_i^Y = \lambda_i^X$ and $\lambda_i^Z = \lambda_i^X - \alpha T$. $\lambda_i^Y = \lambda_i^Z = 0$ otherwise.

Summarizing this argument, we can state that:

$$
\lambda_i^{Y*} = \begin{cases} \mathrm{HT}\left(\lambda_i^X, \alpha T\right), & i \leq \min(k, m) \\ 0, & \text{otherwise} \end{cases}, \qquad
\lambda_i^{Z*} = \begin{cases} \mathrm{ST}\left(\lambda_i^X, \alpha T\right), & i \leq \min(k, m) \\ 0, & \text{otherwise} \end{cases}
$$

$$\text{(S.45)}$$

optimizes the cost (S.44).

To reconstruct $\mathbf{Y}^*$, using (S.43) and (S.45) we rewrite the eigenvalue decomposition of $\mathbf{Y}^{*\top}\mathbf{Y}^*$. Using $\mathbf{\Lambda}^{Y*}$ to denote optimal singular values defined by (S.45), and $\mathbf{O}_{\mathbf{\Lambda}^{Y*}}$ to denote an orthogonal matrix that commutes with $\mathbf{\Lambda}^{Y*}$, we get:

$$
\begin{aligned}
\mathbf{Y}^{*\top}\mathbf{Y}^* &= \mathbf{V}^X\mathbf{O}_{\mathbf{\Lambda}^X}\mathbf{O}_{\mathbf{\Lambda}^{Y*}}\mathbf{\Lambda}^{Y*}\mathbf{O}_{\mathbf{\Lambda}^{Y*}}^\top\mathbf{O}_{\mathbf{\Lambda}^X}^\top\mathbf{V}^{X\top} \\
&= \mathbf{V}^X\mathbf{O}_{\mathbf{\Lambda}^X}\mathbf{\Lambda}^{Y*}\mathbf{O}_{\mathbf{\Lambda}^X}^\top\mathbf{V}^{X\top}.
\end{aligned}
$$

$$\text{(S.46)}$$

But,

$$
\left[\mathbf{\Lambda}^{Y*}, \mathbf{O}_{\mathbf{\Lambda}^X}\right] = 0. \tag{S.47}
$$

since if diagonal elements of $\mathbf{\Lambda}^X$ are degenerate, corresponding diagonal elements of $\mathbf{\Lambda}^{Y*}$ are degenerate. Hence,

$$
\mathbf{Y}^{*\top}\mathbf{Y}^* = \mathbf{V}^X\mathbf{\Lambda}^{Y*}\mathbf{V}^{X\top}. \tag{S.48}
$$

These $\mathbf{Y}^*$ matrices can be constructed as in (S.38): its columns are coordinates in the arbitrarily rotated orthogonal basis spanning the $k$-dimensional principal subspace of $\mathbf{X}\mathbf{X}^\top$.

To reconstruct $\mathbf{Z}^*$, using (S.41) and (S.45) we rewrite the eigenvalue decomposition of $\mathbf{Z}^{*\top}\mathbf{Z}^*$. Using $\mathbf{\Lambda}^{Z*}$ to denote optimal singular values defined by (S.45), and $\mathbf{Q}_{\mathbf{\Lambda}^{Z*}}$ to denote an orthogonal matrix that commutes with $\mathbf{\Lambda}^{Z*}$, we get:

$$\begin{aligned}
\mathbf{Z}^{*\top}\mathbf{Z}^* &= \mathbf{V}^{Y*}\mathbf{Q}_{\mathbf{\Lambda}^{Y*}}\mathbf{Q}_{\mathbf{\Lambda}^{Z*}}\mathbf{\Lambda}^{Z*}\mathbf{Q}_{\mathbf{\Lambda}^{Z*}}^{\top}\mathbf{Q}_{\mathbf{\Lambda}^{Y*}}^{\top}\mathbf{V}^{Y*\top} \\
&= \mathbf{V}^{Y*}\mathbf{Q}_{\mathbf{\Lambda}^{Y*}}\mathbf{\Lambda}^{Z*}\mathbf{Q}_{\mathbf{\Lambda}^{Y*}}^{\top}\mathbf{V}^{Y*\top}.
\end{aligned} \tag{S.49}$$

But,

$$\left[\mathbf{\Lambda}^{Z*}, \mathbf{Q}_{\mathbf{\Lambda}^{Y*}}\right] = 0. \tag{S.50}$$

since if diagonal elements of $\mathbf{\Lambda}^{Y*}$ are degenerate, corresponding diagonal elements of $\mathbf{\Lambda}^{Z*}$ are degenerate . Hence,

$$\mathbf{Z}^{*\top}\mathbf{Z}^* = \mathbf{V}^{Y*}\mathbf{\Lambda}^{Z*}\mathbf{V}^{Y*\top}. \tag{S.51}$$

Plugging in for $\mathbf{V}^{Y*}$, one gets

$$\mathbf{Z}^{*\top}\mathbf{Z}^* = \mathbf{V}^X\mathbf{O}_{\mathbf{\Lambda}^X}\mathbf{O}_{\mathbf{\Lambda}^{Y*}}\mathbf{\Lambda}^{Z*}\mathbf{O}_{\mathbf{\Lambda}^{Y*}}^{\top}\mathbf{O}_{\mathbf{\Lambda}^X}^{\top}\mathbf{V}^{X\top}. \tag{S.52}$$

But,

$$\left[\mathbf{\Lambda}^{Z*}, \mathbf{O}_{\mathbf{\Lambda}^{Y*}}\right] = 0 \tag{S.53}$$

and

$$\left[\mathbf{\Lambda}^{Z*}, \mathbf{O}_{\mathbf{\Lambda}^X}\right] = 0. \tag{S.54}$$

since if diagonal elements of $\mathbf{\Lambda}^{Y*}$ are degenerate, corresponding diagonal elements of $\mathbf{\Lambda}^{Z*}$ are degenerate and if diagonal elements of $\mathbf{\Lambda}^X$ are degenerate, corresponding diagonal elements $\mathbf{\Lambda}^{Z*}$ are degenerate. Then,

$$\mathbf{Z}^{*\top}\mathbf{Z}^* = \mathbf{V}^X\mathbf{\Lambda}^{Z*}\mathbf{V}^{X\top}. \tag{S.55}$$

These $\mathbf{Z}^*$ matrices can be constructed as in (S.38). $\qquad\square$

*Remark* 2A. Here we comment on the case $l < \min(k, m)$. In the eigenvalue cost (S.44), among the terms for which $\lambda_i^X > \alpha T$, there will be cases where $\lambda_i^Z$ is forced to be zero, while $\lambda_i^Y \geq 0$. The cost for such terms are $\lambda_i^{X2} - \alpha^2 T^2 - 2\lambda_i^Y\left(\lambda_i^X - \alpha T\right)$, which minimizes when $\lambda_i^Y \to \infty$. We found through numerical simulations that the corresponding online algorithm is unstable in this regime.

*Remark* 2B. Here we comment on the case where $\alpha$ is an eigenvalue of $\mathbf{C}$. Here we need to consider optimization of terms for which $\lambda_i^X = \alpha T$ in the eigenvalue cost (S.44). The cost for such terms are $2\lambda_i^Z \left( \lambda_i^Y - \alpha T \right) - \lambda_i^{Z 2}$, which is optimized for any $\lambda_i^Y \leq \alpha T$ and $\lambda_i^Z = 0$ with a 0 value for the cost. To see this, consider the other case $\lambda_i^Y > \alpha T$. Then the optimization with respect to $\lambda_i^Z$ would give $\lambda_i^Z = \lambda_i^Y - \alpha T$ and the cost of the term would be $\left( \lambda_i^Y - \alpha T \right)^2 > 0$, which would be suboptimal. Hence, if $k \geq m$, or $k < m$ and $\lambda_k^X = \alpha T$, $\mathbf{Y} \in \mathbb{R}^{k \times T}$ and $\mathbf{Z} \in \mathbb{R}^{l \times T}$ constructed as in (S.38) are still optimal, however there are other optimal solutions. Non-zero $\{\lambda_i^Y\}$ corresponding to $\alpha$ eigenvalue of $\mathbf{C}$ can take values $0 \leq \lambda_i^Y \leq \alpha T$.

*Remark* 2C. Here we comment on the case $k < m$ and $\lambda_k^X = \lambda_{k+1}^X$. We first discuss how optimal $\mathbf{Y}$ change. In this case, (S.47) is not generally true anymore, because while $\lambda_k^X = \lambda_{k+1}^X$, $\lambda_k^Y \neq \lambda_{k+1}^Y$. $\mathbf{Y}$ matrices constructed as in (S.38) are still minima, as can be seen by choosing $\mathbf{O}_{\mathbf{\Lambda}^X} = \mathbf{I}_T$ for which (S.47) holds, but there are other solutions which cannot be put in the form (S.38). We observe that blocks of $\mathbf{O}_{\mathbf{\Lambda}^X}$ that do not correspond to $\lambda_k^X$ still commute with $\mathbf{\Lambda}^{Y*}$. Thus, when $k < m$ and $\lambda_k^X = \lambda_{k+1}^X$, we can write the most general solution as

$$\mathbf{Y}^* = \mathbf{U}_k \, \mathbf{\Lambda}_{k \times T}^{Y*}{}^{1/2} \, \mathbf{O}_{\Lambda_k^X}^\top \mathbf{V}^{X\top}, \tag{S.56}$$

where $\mathbf{U}_k$ is a $k \times k$ orthogonal matrix, $\mathbf{\Lambda}_{k \times T}^{Y*}$ is a $k \times T$ diagonal matrix with its $k$ diagonals set to first $k$ diagonals of $\mathbf{\Lambda}^{Y*}$, and $\mathbf{O}_{\Lambda_k^X}$ is a $T \times T$ orthogonal matrix that is diagonal except one block that corresponds to diagonal elements of $\mathbf{\Lambda}^X$ that are degenerate with $\lambda_k^X$. Next we discuss how optimal $\mathbf{Z}$ change. In this case, while (S.53) is still true, (S.54) is not generally true anymore, because while $\lambda_k^X = \lambda_{k+1}^X$, $\lambda_k^Z \neq \lambda_{k+1}^Z$. $\mathbf{Z}$ matrices constructed as in (S.38) are still minima, as can be seen by choosing $\mathbf{O}_{\mathbf{\Lambda}^X} = \mathbf{I}_T$ for which (S.54) holds, but there are other solutions which cannot be put in the form (S.38). We observe that blocks of $\mathbf{O}_{\mathbf{\Lambda}^X}$ that do not correspond to $\lambda_k^X$ still commute with $\mathbf{\Lambda}^{Z*}$. Thus, when $k < m$ and $\lambda_k^X = \lambda_{k+1}^X$, we can write the most general solution as

$$\mathbf{Z}^* = \mathbf{U}_l \, \mathbf{\Lambda}_{l \times T}^{Z*}{}^{1/2} \, \mathbf{O}_{\Lambda_k^X}^\top \mathbf{V}^{X\top}, \tag{S.57}$$

where $\mathbf{U}_l$ is an $l \times l$ orthogonal matrix, $\mathbf{\Lambda}_{l \times T}^{Z*}$ is a $l \times T$ diagonal matrix with its $l$ diagonals set to first $l$ diagonals of $\mathbf{\Lambda}^{Z*}$, and $\mathbf{O}_{\Lambda_l^X}$ is a $T \times T$ orthogonal matrix that is diagonal except one block that corresponds to diagonal elements of $\mathbf{\Lambda}^X$ that are degenerate with $\lambda_k^X$.

## IV.   PROOF OF THEOREM 3 - THRESHOLDING AND EQUALIZATION OF COVARIANCE EIGENVALUES

We reproduce the offline objective (8) for ease of referencing:

$$\min_{\mathbf{Y}} \max_{\mathbf{Z}} \operatorname{Tr} \left[ -\mathbf{X}^\top \mathbf{X} \mathbf{Y}^\top \mathbf{Y} + \mathbf{Y}^\top \mathbf{Y} \mathbf{Z}^\top \mathbf{Z} + \alpha T \mathbf{Y}^\top \mathbf{Y} - \beta T \mathbf{Z}^\top \mathbf{Z} \right], \qquad \text{(S.58)}$$

where $\alpha \geq 0$ and $\beta \geq 0$. Let $m$ be the number of eigenvalues $\mathbf{C}$ greater than $\alpha$.

**Theorem 3.** *Suppose an eigen-decomposition of $\mathbf{X}^\top \mathbf{X}$ is $\mathbf{X}^\top \mathbf{X} = \mathbf{V}^X \mathbf{\Lambda}^X \mathbf{V}^{X^\top}$, where $\mathbf{\Lambda}^X = \operatorname{diag}\left( \lambda_1^X, \ldots, \lambda_T^X \right)$ with $\lambda_1^X \geq \ldots \geq \lambda_T^X \geq 0$. Assume $l \geq \min(k, m)$. Then,*

$$\mathbf{Y}^* = \mathbf{U}_k \sqrt{\beta T} \, \mathbf{\Theta}_k(\mathbf{\Lambda}^X, \alpha T)^{1/2} \, \mathbf{V}_k^{X^\top}, \qquad \mathbf{Z}^* = \mathbf{U}_l \, \mathbf{\Sigma}_{l \times T} \mathbf{O}_{\mathbf{\Lambda}^{Y*}} \mathbf{V}^{X^\top}, \qquad \text{(S.59)}$$

*are optima of (S.58), where $\mathbf{\Theta}_k(\mathbf{\Lambda}^X, \alpha T) = \operatorname{diag}\left( \Theta\left( \lambda_1^X - \alpha T \right), \ldots, \Theta\left( \lambda_k^X - \alpha T \right) \right)$, $\mathbf{\Sigma}_{l \times T}$ is an $l \times T$ rectangular diagonal matrix with top $\min(k, m)$ diagonals are set to arbitrary nonnegative constants and the rest are zero, $\mathbf{O}_{\mathbf{\Lambda}^{Y*}}$ is a block-diagonal orthogonal matrix that has two blocks: the top block is $\min(k, m)$ dimensional and the bottom block is $T - \min(k, m)$ dimensional, $\mathbf{V}_p = \left[ \mathbf{v}_1^X, \ldots, \mathbf{v}_p^X \right]$, and $\mathbf{U}_p \in O(p)$. The form (S.59) uniquely defines all optima of (S.58) except when either 1) $\alpha$ is an eigenvalue of $\mathbf{C}$ or 2) $k < m$ and $\lambda_k^X = \lambda_{k+1}^X$.*

*Proof.* Here we assume 1) $l \geq \min(k, m)$, 2) $\alpha$ is not an eigenvalue of $\mathbf{C}$ and 3) if $k < m$, then $\lambda_k^X \neq \lambda_{k+1}^X$. We prove that with these assumptions, the form (S.59) uniquely defines all optima (S.58). Violations of these assumptions are treated in three remarks below.

The proof is similar to that of Theorem 1. Since the cost (S.58) depends on $\mathbf{Y}$ only through the similarity matrix $\mathbf{Y}^\top \mathbf{Y}$ and $\mathbf{Z}$ through $\mathbf{Z}^\top \mathbf{Z}$, we find optimizing $\mathbf{Y}^\top \mathbf{Y}$ and $\mathbf{Z}^\top \mathbf{Z}$ from which we reconstruct $\mathbf{Y}$ and $\mathbf{Z}$. Our strategy is to start with eigendecompositions of $\mathbf{Y}^\top \mathbf{Y} = \mathbf{V}^Y \mathbf{\Lambda}^Y \mathbf{V}^{Y^\top}$ and $\mathbf{Z}^\top \mathbf{Z} = \mathbf{V}^Z \mathbf{\Lambda}^Z \mathbf{V}^{Z^\top}$, and find optimal $\mathbf{V}^Y$, $\mathbf{\Lambda}^Y$, $\mathbf{V}^Z$ and $\mathbf{\Lambda}^Z$.

We first optimize for $\mathbf{V}^Y \in \mathbb{R}^{T \times T}$ and $\mathbf{V}^Z \in \mathbb{R}^{T \times T}$ for fixed $\mathbf{\Lambda}^Y$ and $\mathbf{\Lambda}^Z$. Because Frobenius norm is invariant under rotations, the terms in objective (S.59) can be rewritten as

$$\operatorname{Tr} \left[ - \left( \mathbf{\Lambda}^X - \alpha T \mathbf{I}_T \right) \mathbf{O} \mathbf{\Lambda}^Y \mathbf{O}^\top + \left( \mathbf{\Lambda}^Y - \beta T \mathbf{I}_T \right) \mathbf{Q} \mathbf{\Lambda}^Z \mathbf{Q}^\top \right], \qquad \text{(S.60)}$$

where $\mathbf{O} = \mathbf{V}^{X^\top} \mathbf{V}^Y \in O(T)$ and where $\mathbf{Q} = \mathbf{V}^{Y^\top} \mathbf{V}^Z \in O(T)$. First, we do the maximization over $\mathbf{V}^Z$, which entails the second term of (S.60). According to Lemma 2, all orthogonal

matrices $\mathbf{Q} = \mathbf{V}^{Y\top}\mathbf{V}^Z$ that are a product of two orthogonal matrices $\mathbf{Q} = \mathbf{Q}_{\mathbf{\Lambda}^Y}\mathbf{Q}_{\mathbf{\Lambda}^Z}$ with $[\mathbf{\Lambda}^Y - \beta T\mathbf{I}_T, \mathbf{Q}_{\mathbf{\Lambda}^Y}] = 0$ and $[\mathbf{\Lambda}^Z, \mathbf{Q}_{\mathbf{\Lambda}^Z}] = 0$, are optimal. Then, optimal $\mathbf{V}^Z$ are given by

$$\mathbf{V}^{Z*} = \mathbf{V}^Y\mathbf{Q}_{\mathbf{\Lambda}^Y}\mathbf{Q}_{\mathbf{\Lambda}^Z}. \tag{S.61}$$

For this choice of $\mathbf{V}^Z$, the second term in (S.60) is $\text{Tr}\left((\mathbf{\Lambda}^Y - \beta T\mathbf{I}_T)\mathbf{\Lambda}^Z\right)$ and therefore the minimization over $\mathbf{V}^Y$ only entails the first term in (S.60). According to Lemma 2, all orthogonal matrices $\mathbf{O} = \mathbf{V}^{X\top}\mathbf{V}^Y$ that are a product of two orthogonal matrices $\mathbf{O} = \mathbf{O}_{\mathbf{\Lambda}^X}\mathbf{O}_{\mathbf{\Lambda}^Y}$ with $[\mathbf{\Lambda}^X, \mathbf{O}_{\mathbf{\Lambda}^X}] = 0$ and $[\mathbf{\Lambda}^Y, \mathbf{O}_{\mathbf{\Lambda}^Y}] = 0$, are optimal. Then, optimal $\mathbf{V}^Y$ are given by

$$\mathbf{V}^{Y*} = \mathbf{V}^X\mathbf{O}_{\mathbf{\Lambda}^X}\mathbf{O}_{\mathbf{\Lambda}^Y}. \tag{S.62}$$

For these choices of $\mathbf{V}^{Y*}$ and $\mathbf{V}^{Z*}$, the full objective (S.58) reduces to:

$$\min_{\lambda_1^Y,\ldots,\lambda_T^Y} \max_{\lambda_1^Z,\ldots,\lambda_T^Z} \sum_{i=1}^T \left[-\left(\lambda_i^X - \alpha T\right)\lambda_i^Y + \left(\lambda_i^Y - \beta T\right)\lambda_i^Z\right], \tag{S.63}$$

where $\{\lambda_1^Z,\ldots,\lambda_T^Z\}$ are constrained to be non-negative and at most $l$ of them are non-zero, and $\{\lambda_1^Y,\ldots,\lambda_T^Y\}$ are also constrained to be non-negative and at most $k$ of them are non-zero. We analyze the terms in the sum separately:

1. Consider the $i > m$ terms in the sum for which $\lambda_i^X < \alpha T$. For such terms, choosing $\lambda_i^Y = \lambda_i^Z = 0$ gives the optimal cost, 0. To see this, let's calculate costs associated with other choices of $\lambda_i^Y$ and $\lambda_i^Z$. Suppose $\lambda_i^Y > \beta T$. Then, maximization with respect to $\lambda_i^Z$ would set the cost to $\infty$. Suppose $\lambda_i^Y = \beta T$. Then, the coefficient in front of $\lambda_i^Z$ is 0, and the cost is $-\left(\lambda_i^X - \alpha T\right)\beta T > 0$. Suppose $\lambda_i^Y < \beta T$. Then, maximization with respect to $\lambda_i^Z$ would set $\lambda_i^Z = 0$ and the cost is $-\left(\lambda_i^X - \alpha T\right)\lambda_i^Y$, which is minimal at $\lambda_i^Y = 0$. Hence, our claim holds.

2. Consider the $i \leq m$ terms in the sum for which $\lambda_i^X > \alpha T$. Note that we assumed $\alpha$ is not an eigenvalue of $\mathbf{C}$, therefore we omit the equality case. Since $\lambda_{i>m}^Y = \lambda_{i>m}^Z = 0$, and we assumed $l \geq m$, we can assign all $\lambda_{i\leq m}^Z$ to non-zero values if needed. On the other hand, $k$ can be less than $m$ and we might be forced to set some $\lambda_{i\leq m}^Y$ to zero.

   (a) Suppose $\lambda_i^Y > 0$. For these terms, choosing $\lambda_i^Y = \beta T$ and any $\lambda_i^Z$ gives the optimal cost, $-\left(\lambda_i^X - \alpha T\right)\beta T < 0$. To see this, let's calculate costs associated

with other choices of $\lambda_i^Y$ and $\lambda_i^Z$. If $\lambda_i^Y > \beta T$, maximization with respect to $\lambda_i^Z$ would set the cost to $\infty$. If $\lambda_i^Y < \beta T$, maximization with respect to $\lambda_i^Z$ would set $\lambda_i^Z = 0$, and the cost would be $-\left(\lambda_i^X - \alpha T\right)\lambda_i^Y$, which is greater than its value at $\lambda_i^Y = \beta T$, given by $-\left(\lambda_i^X - \alpha T\right)\beta T$. Hence, our claim holds.

(b) Suppose $\lambda_i^Y = 0$. For these terms, choosing $\lambda_i^Z = 0$ gives the optimal cost, 0.

Therefore, one assigns non-zero $\lambda_i^Y$ to the the first $\min(k, m)$ terms in the sum. For such terms $\lambda_i^Y = \beta T$ and $\lambda_i^Z$ can take any value.

Summarizing this argument, we can state that:

$$\lambda_i^{Y*} = \begin{cases} \beta T, & i \le \min(k,m) \\ 0, & \text{otherwise} \end{cases}, \qquad \lambda_i^{Z*} = \begin{cases} \text{any non-negative value}, & i \le \min(k,m) \\ 0, & \text{otherwise} \end{cases}$$

(S.64)

optimizes the cost (S.63).

To reconstruct $\mathbf{Y}^*$, using (S.62) and (S.64) we rewrite the eigenvalue decomposition of $\mathbf{Y}^{*\top}\mathbf{Y}^*$. Using $\mathbf{\Lambda}^{Y*}$ to denote optimal singular values defined by (S.64), and $\mathbf{O}_{\mathbf{\Lambda}^{Y*}}$ to denote an orthogonal matrix that commutes with $\mathbf{\Lambda}^{Y*}$, we get:

$$\begin{aligned}\mathbf{Y}^{*\top}\mathbf{Y}^* &= \mathbf{V}^X \mathbf{O}_{\mathbf{\Lambda}^X}\mathbf{O}_{\mathbf{\Lambda}^{Y*}}\mathbf{\Lambda}^{Y*}\mathbf{O}_{\mathbf{\Lambda}^{Y*}}^\top \mathbf{O}_{\mathbf{\Lambda}^X}^\top \mathbf{V}^{X\top} \\ &= \mathbf{V}^X \mathbf{O}_{\mathbf{\Lambda}^X}\mathbf{\Lambda}^{Y*}\mathbf{O}_{\mathbf{\Lambda}^X}^\top \mathbf{V}^{X\top}.\end{aligned}$$

(S.65)

But,

$$\left[\mathbf{\Lambda}^{Y*}, \mathbf{O}_{\mathbf{\Lambda}^X}\right] = 0.$$

(S.66)

since if diagonal elements of $\mathbf{\Lambda}^X$ are degenerate, corresponding diagonal elements of $\mathbf{\Lambda}^{Y*}$ are degenerate. Hence,

$$\mathbf{Y}^{*\top}\mathbf{Y}^* = \mathbf{V}^X \mathbf{\Lambda}^{Y*}\mathbf{V}^{X\top}.$$

(S.67)

These $\mathbf{Y}^*$ matrices can be constructed as in (S.59): its columns are coordinates in the arbitrarily rotated orthogonal basis spanning the $k$-dimensional principal subspace of $\mathbf{X}\mathbf{X}^\top$.

To reconstruct $\mathbf{Z}^*$, using (S.61) and (S.64) we rewrite the eigenvalue decomposition of $\mathbf{Z}^{*\top}\mathbf{Z}^*$. Using $\mathbf{\Lambda}^{Z*}$ to denote optimal singular values defined by (S.64), and $\mathbf{Q}_{\mathbf{\Lambda}^{Z*}}$ to denote

an orthogonal matrix that commutes with $\mathbf{\Lambda}^{Z*}$, we get:

$$\mathbf{Z}^{*\top}\mathbf{Z}^* = \mathbf{V}^{Y*}\mathbf{Q}_{\mathbf{\Lambda}^{Y*}}\mathbf{Q}_{\mathbf{\Lambda}^{Z*}}\mathbf{\Lambda}^{Z*}\mathbf{Q}_{\mathbf{\Lambda}^{Z*}}^{\top}\mathbf{Q}_{\mathbf{\Lambda}^{Y*}}^{\top}\mathbf{V}^{Y*\top}$$

$$= \mathbf{V}^{Y*}\mathbf{Q}_{\mathbf{\Lambda}^{Y*}}\mathbf{\Lambda}^{Z*}\mathbf{Q}_{\mathbf{\Lambda}^{Y*}}^{\top}\mathbf{V}^{Y*\top}. \tag{S.68}$$

Unlike before, $\left[\mathbf{\Lambda}^{Z*}, \mathbf{Q}_{\mathbf{\Lambda}^{Y*}}\right] \neq 0$ in general. Plugging in for $\mathbf{V}^{Y*}$, we get:

$$\mathbf{Z}^{*\top}\mathbf{Z}^* = \mathbf{V}^X\mathbf{O}_{\mathbf{\Lambda}^X}\mathbf{O}_{\mathbf{\Lambda}^{Y*}}\mathbf{Q}_{\mathbf{\Lambda}^{Y*}}\mathbf{\Lambda}^{Z*}\mathbf{Q}_{\mathbf{\Lambda}^{Y*}}^{\top}\mathbf{O}_{\mathbf{\Lambda}^{Y*}}^{\top}\mathbf{O}_{\mathbf{\Lambda}^X}^{\top}\mathbf{V}^{X\top}. \tag{S.69}$$

This expression can be simplified further. Remembering that $\left[\mathbf{\Lambda}^{Y*}, \mathbf{O}_{\mathbf{\Lambda}^X}\right] = 0$ from (S.66), we can absorb the product $\mathbf{O}_{\mathbf{\Lambda}^X}\mathbf{O}_{\mathbf{\Lambda}^{Y*}}\mathbf{Q}_{\mathbf{\Lambda}^{Y*}}$ into $\mathbf{O}_{\mathbf{\Lambda}^{Y*}}$, a single orthogonal matrix that commutes with $\mathbf{\Lambda}^{Y*}$,

$$\mathbf{Z}^{*\top}\mathbf{Z}^* = \mathbf{V}^X\mathbf{O}_{\mathbf{\Lambda}^{Y*}}\mathbf{\Lambda}^{Z*}\mathbf{O}_{\mathbf{\Lambda}^{Y*}}^{\top}\mathbf{V}^{X\top}. \tag{S.70}$$

What is the structure of $\mathbf{O}_{\mathbf{\Lambda}^{Y*}}$? Since $\mathbf{\Lambda}^{Y*}$ has top $\min(k, m)$ diagonals $\beta T$ and rest zero, (S.64), $\mathbf{O}_{\mathbf{\Lambda}^{Y*}}$ has two blocks, first is $\min(k, m)$ dimensional and the second is $T - \min(k, m)$ dimensional. These $\mathbf{Z}^*$ matrices can be constructed as in (S.38).

$\square$

*Remark* 3A. Here we comment on the case $l < \min(k, m)$. In the eigenvalue cost (S.63), among the terms for which $\lambda_i^X > \alpha T$, there will be cases where $\lambda_i^Z$ is forced to be zero, while $\lambda_i^Y \geq 0$. The cost for such terms are $-\left(\lambda_i^X - \alpha T\right)\lambda_i^Y$, which minimizes when $\lambda_i^Y \to \infty$. We found through numerical simulations that the corresponding online algorithm is unstable in this regime.

*Remark* 3B. Here we comment on the case where $\alpha$ is an eigenvalue of $\mathbf{C}$. Here we need to consider optimization of terms for which $\lambda_i^X = \alpha T$ in the eigenvalue cost (S.63). The cost for such terms are $\left(\lambda_i^Y - \beta T\right)\lambda_i^Z$, which is optimized for any $\lambda_i^Y \leq \beta T$ and $\lambda_i^Z = 0$ with a $0$ value for the cost. To see this, consider the other case $\lambda_i^Y > \beta T$. Then the optimization with respect to $\lambda_i^Z$ would give $\infty$ cost. Hence, if $k > m$, or $k < m$ and $\lambda_k^X = \alpha T$, $\mathbf{Y} \in \mathbb{R}^{k \times T}$ and $\mathbf{Z} \in \mathbb{R}^{l \times T}$ constructed as in (S.59) are still optimal, however there are other optimal solutions. Non-zero $\{\lambda_i^Y\}$ corresponding to $\alpha$ eigenvalue of $\mathbf{C}$ can take values $0 \leq \lambda_i^Y \leq \alpha T$.

*Remark* 3C. Here we comment on the case $k < m$ and $\lambda_k^X = \lambda_{k+1}^X$. We first discuss how optimal $\mathbf{Y}$ change. In this case, (S.66) is not generally true anymore, because while $\lambda_k^X = \lambda_{k+1}^X$, $\lambda_k^Y \neq \lambda_{k+1}^Y$. $\mathbf{Y}$ matrices constructed as in (S.59) are still minima, as can be seen by

choosing $\mathbf{O}_{\boldsymbol{\Lambda}^X} = \mathbf{I}_T$ for which (S.66) holds, but there are other solutions which cannot be put in the form (S.59). We observe that blocks of $\mathbf{O}_{\boldsymbol{\Lambda}^X}$ that do not correspond to $\lambda_k^X$ still commute with $\boldsymbol{\Lambda}^{Y*}$. Thus, when $k < m$ and $\lambda_k^X = \lambda_{k+1}^X$, we can write the most general solution as

$$\mathbf{Y}^* = \mathbf{U}_k \, \boldsymbol{\Lambda}_{k \times T}^{Y*}{}^{1/2} \, \mathbf{O}_{\Lambda_k^X}^\top \mathbf{V}^{X\top},  \tag{S.71}$$

where $\mathbf{U}_k$ is a $k \times k$ orthogonal matrix, $\boldsymbol{\Lambda}_{k \times T}^{Y*}$ is a $k \times T$ diagonal matrix with its $k$ diagonals set to first $k$ diagonals of $\boldsymbol{\Lambda}^{Y*}$, and $\mathbf{O}_{\Lambda_k^X}$ is a $T \times T$ orthogonal matrix that is diagonal except one block that corresponds to diagonal elements of $\boldsymbol{\Lambda}^X$ that are degenerate with $\lambda_k^X$. Next we discuss how optimal $\mathbf{Z}$ change. In this case, while (S.69) is still true, (S.70) is not generally true anymore, because (S.66) does not hold in general. $\mathbf{Z}$ matrices constructed as in (S.59) are still minima, as can be seen by choosing $\mathbf{O}_{\boldsymbol{\Lambda}^X} = \mathbf{I}_T$ for which (S.70) holds, but there are other solutions which cannot be put in the form (S.59). We observe that blocks of $\mathbf{O}_{\boldsymbol{\Lambda}^X}$ that do not correspond to $\lambda_k^X$ still commute with $\boldsymbol{\Lambda}^{Y*}$. Thus, when $k < m$ and $\lambda_k^X = \lambda_{k+1}^X$, we can write the most general solution as

$$\mathbf{Z}^* = \mathbf{U}_l \, \boldsymbol{\Lambda}_{l \times T}^{Z*}{}^{1/2} \, \mathbf{O}_{\Lambda_k^X}^\top \mathbf{O}_{\Lambda^{Y*}}^\top \mathbf{V}^{X\top},  \tag{S.72}$$

where $\mathbf{U}_l$ is an $l \times l$ orthogonal matrix, $\boldsymbol{\Lambda}_{l \times T}^{Z*}$ is a $l \times T$ diagonal matrix with its $l$ diagonals set to first $l$ diagonals of $\boldsymbol{\Lambda}^{Z*}$, and $\mathbf{O}_{\Lambda_k^X}$ is a $T \times T$ orthogonal matrix that is diagonal except one block that corresponds to diagonal elements of $\boldsymbol{\Lambda}^X$ that are degenerate with $\lambda_k^X$.

## V.   FULL EXPRESSIONS FOR INPUT-TO-OUPUT MAPPING MATRICES

Here we give the full expressions for linear transformation that maps inputs, $\mathbf{x}_T$, to outputs,

$$\mathbf{y}_T = \mathbf{F}_T^{YX} \mathbf{x}_T, \qquad \mathbf{z}_T = \mathbf{F}_T^{ZX} \mathbf{x}_T.  \tag{S.73}$$

To do this, we find fixed points of the neural dynamics stages of the three algorithms.

### A.   Online soft-thresholding of eigenvalues

The neural dynamics stage of this algorithm is

$$\mathbf{y}_T \leftarrow (1 - \eta) \, \mathbf{y}_T + \eta \left( \mathbf{W}_T^{YX} \mathbf{x}_T - \mathbf{W}_T^{YY} \mathbf{y}_T \right).  \tag{S.74}$$

At the fixed point of this iteration,

$$\left(\mathbf{I}_m + \mathbf{W}_T^{YY}\right) \mathbf{y}_T = \mathbf{W}_T^{YX} \mathbf{x}_T, \tag{S.75}$$

and therefore

$$\mathbf{F}_T^{YX} = \left(\mathbf{I}_m + \mathbf{W}_T^{YY}\right)^{-1} \mathbf{W}_T^{YX}. \tag{S.76}$$

### B.    Online hard-thresholding of eigenvalues

The neural dynamics stage of this algorithm is

$$\mathbf{y}_T \leftarrow (1 - \eta) \, \mathbf{y}_T + \eta \left(\mathbf{W}_T^{YX} \mathbf{x}_T - \mathbf{W}_T^{YZ} \mathbf{z}_T\right),$$
$$\mathbf{z}_T \leftarrow (1 - \eta) \, \mathbf{z}_T + \eta \left(\mathbf{W}_T^{ZY} \mathbf{y}_T - \mathbf{W}_T^{ZZ} \mathbf{z}_T\right). \tag{S.77}$$

At the fixed point of this iteration,

$$\mathbf{y}_T = \mathbf{W}_T^{YX} \mathbf{x}_T - \mathbf{W}_T^{YZ} \mathbf{z}_T,$$
$$\left(\mathbf{I}_k + \mathbf{W}_T^{ZZ}\right) \mathbf{z}_T = \mathbf{W}_T^{ZY} \mathbf{y}_T, \tag{S.78}$$

and therefore

$$\mathbf{F}_T^{YX} = \left(\mathbf{I}_m + \mathbf{W}^{YZ} \left(\mathbf{I}_k + \mathbf{W}_T^{ZZ}\right)^{-1} \mathbf{W}^{ZY}\right)^{-1} \mathbf{W}_T^{YX},$$
$$\mathbf{F}_T^{ZX} = \left(\mathbf{I}_k + \mathbf{W}_T^{ZZ}\right)^{-1} \mathbf{W}_T^{ZY} \mathbf{F}_T^{YX}. \tag{S.79}$$

### C.    Online thresholding and equalization of eigenvalues

The neural dynamics stage of this algorithm is

$$\mathbf{y}_T \leftarrow (1 - \eta) \, \mathbf{y}_T + \eta \left(\mathbf{W}_T^{YX} \mathbf{x}_T - \mathbf{W}_T^{YZ} \mathbf{z}_T\right),$$
$$\mathbf{z}_T \leftarrow (1 - \eta) \, \mathbf{z}_T + \eta \mathbf{W}_T^{ZY} \mathbf{y}_T. \tag{S.80}$$

At the fixed point of this iteration,

$$\mathbf{y}_T = \mathbf{W}_T^{YX} \mathbf{x}_T - \mathbf{W}_T^{YZ} \mathbf{z}_T,$$
$$\mathbf{z}_T = \mathbf{W}_T^{ZY} \mathbf{y}_T, \tag{S.81}$$

and therefore

$$\mathbf{F}_T^{YX} = \left(\mathbf{I}_m + \mathbf{W}^{YZ}\mathbf{W}^{ZY}\right)^{-1}\mathbf{W}_T^{YX},$$

$$\mathbf{F}_T^{ZX} = \mathbf{W}_T^{ZY}\mathbf{F}_T^{YX}. \tag{S.82}$$

[1] R. A. Horn and C. R. Johnson, *Matrix analysis* (Cambridge university press, 2012).