[Reviews · NeurIPS 2015]

Submitted by Assigned_Reviewer_1

### Summary A normative approach is presented to derive local learning rules for a rate network that optimise a classic multidimensional scaling objective (MDS). The work extends a recent publication on the same topic in three ways: (1) it incorporates a low-rank regularisation on the network output, (2) adds another set of latent variables to match the distribution of Hebbian/Anti-Hebbian plasticity rules in cortical networks and (3) adds a penalty term to whiten the outputs.

### Advances The manuscript represents a welcome addition to the (recently) growing set of publications that strive to derive local learning rules in a normative way. The ansatz presented here is interesting, though much of the novelty is already presented in the earlier publication [22]. The additions are nonetheless big enough to justify another publication. For one, it had so far been unclear whether low-rank regularisations can actually be incorporated into local learning rules, and the work here sets a nice precedence. Second, anti-hebbian rules are usually observed on inhibitory synapses, and so the extension by another set of latent variables with similar characteristics is welcome.

### Major First and foremost, the paper is not the first one to derive biologically plausible learning rule from a normative ansatz (as suggested in line 065). Besides the earlier publication on which this manuscript is build upon, there is at least the publication by Vertechi et al. (2014) [1] which derive neural networks and local learning rules for a rate-based neural networks that converges to an efficient population code (for both the over- as well as under complete case, the latter of which performs dimensionality reduction).

The objectives (3) and (5) are difficult to understand in the way written here, and should be rather formulated in terms of the standard MDS with an additional nuclear norm regularisation on Y^TY. E. g. (3) is equivalent to

min_Y ||X^TX - Y^TY||_2^2 + alpha*T*||Y^TY||_*

where ||.||_* is the nuclear norm (i.e. the sum of singular values). The authors note this relation in passing, but it would really help the reader if the intuition behind (3) and (5) is explained and formulated more extensively.

More severely, I could not develop any intuition behind objective (7). This equation is essentially (5) without the quadratic terms, but there are no explanations whatsoever as to where this objective is coming from, what it is supposed to capture, and what it adds to the objectives (3) and (5).

### Minor 1) Line 039 "each individual neuron still has to compress hundreds or thousands of its inputs into a single output". I do not doubt the importance of compression for the brain, but that is a very odd argument given that it is true for any linear transformation (being it expansive or contractive). 2) Line 044 "Because the signal-to-noise ratio of natural stimuli varies significantly [...]". I have no idea what that should mean - are you referring to measurement noise like in the retina? 3) Practical problems: how do you initialise the network?

### Verdict The manuscript provides an interesting extension of a recent work on multidimensional scaling in plausible neural networks that can incorporate some rough statistics of synaptic plasticity as well as low-rank approximations using local learning rules alone. The manuscript could gain from more polishing, especially by providing intuition on the different objectives (3), (5) and (7).

[1] Vertechi, Brendel & Machines, Unsupervised learning of an efficient short-term memory network, NIPS (2014)

Update: I have seen the (helpful) response of the authors and I am confident that the points raised above will be sufficiently addressed during the revision.
Summary: The manuscript presents a solid attempt to extend a recent publication on distributed optimisation of a Multidimensional Scaling objective in a neural network, and includes a low-rank constraint, another set of latent variables (alluding to inhibitory neurons) and whitened outputs.

Submitted by Assigned_Reviewer_2

I enjoyed reading this paper, which I think is very clearly structured and written with the right level of details. Minor clarity concern: perhaps the _adaptive_ nature of the dim. reduction presented here could be better spelled out, e.g. by more explicitly pointing out how specifying a _threshold_ on the eigenvalues is different from specifying a fixed number of modes.

Minor wrinkle 1: you perform dimensionality reduction without (necessarily) reducing the number of neurons; instead, you project the input onto a lower dimensional subspace within some higher-dimensional output space (number of output neurons), which can be interpreted as instating (just the right amount of) redundancy. This is quite different from dimensionality reduction techniques that actually do reduce the physical dimension (number of neurons) and therefore provide outputs that cost much less per bit of information (which, btw, is the observation you use in the very first paragraph of your intro). I suppose there is a tradeoff here between energy-efficiency and robustness to output noise; may I suggest you comment on that even briefly?

Minor wrinkle 2: the fact that adaptive dimensionality is achieved by setting an absolute threshold on the eigenvalues (true for each of the 3 schemes you propose) means that you need an a priori knowledge of the level of noise in the input, so you can set your threshold to, say, 3 times that noise variance. Ideally though, it would be nice to have an adaptive algorithm that is instead parametrized by a relative threshold ("I want the circuit to retain 90% of the input variance"). Can you also briefly comment on that?

l38: I think the argument that each neuron compresses its signal down to a one-dimensional output has little to do with the problem at hand; it's a bit like saying that doing Ax (where A is square, dense, and full-rank) is a dimensionality reduction operation, on the basis that every element of Ax is a one-dimensional average of the entries of x... The argument is just a bit odd especially at the end of this paragraph.

typos: 46: dimensions 129: algorithms

Update: I've seen the authors' response and I am sticking to my original evaluation.
Summary: This paper presents a couple of very clever results related to the neural implementation of dimensionality reduction, which significantly extend old-school Oja/Foldiak type of results. Despite a couple of wrinkles, I think it's going to have some impact in theoretical neuroscience especially if it turns out it can be extended to nonlinear, non-Gaussian cases (although a neuroscientist would be hard-pressed to find any biological relevance to the model, in the context of NIPS it seems to be as neurally plausible as it gets).

Submitted by Assigned_Reviewer_3

The author(s) suggested meta-principles of dimensionality reduction of neural network, which may create learning rules. The learning rules obtained from these meta-rule include Hebbian or anti-Hebbian rules, and networks generated under these learning rules turned out to be similar to existing biological networks. It appears interesting to see such production process, but I am not positive for discussing such meta-principles, because the learning principles are not general; different principles corresponded to different biological networks. It is relatively easy to invent meta principles for a particular existence in a posthoc manner, but this brings no universal prediction with regard to the other networks.
Summary: The author(s) suggested principles of dimensionality reduction of neural network, which may create learning rules. The obtained learning rule generated networks similar to the existing biological systems. Though interesting, I do not see scientific meaning of searching for the organization principle of this kind, with regard to the generality.

Submitted by Assigned_Reviewer_4

The statement in the abstract that "a normativ theory of neural dimensionality reduction does not exist" is too strong. Certainly, this problem has been considered by many authors include Wainwright 1999, Fairhall et al. 2001, Brenner et al 2000, Gepshtein et al. J Vision 2007

have all proposed normative theories of adaptation.

The example of dimensionality reduction in the retina from photoreceptors to ganglion cells is not convincing because of differences in temporal resolution between receptors and RGC, not to mention that majority of photoreceptors are rods.

Summary: This paper presents both offline and online rules for

performing dimensionality reduction by adaptive adjusting the number of active neurons. A cause of concern that dimensionality is defined as the number of active output neurons. There are two concerns about this measure. First, it is not clear on what time scale activity of neurons is measured, and whether the dimensionality will fluctuate rapidly with the stimulus. Second, the chosen measure of dimensionality does not take into account correlations in neuronal output. The second concern is perhaps partly mitigated by soft-thresholding implemented in the paper, because small eigenvalues are set to zero. A discussion of these issues would be helpful.

The authors' rebuttal was helpful and I increased my score to "good paper, accept".

Author Feedback
Author rebuttal: We thank all the reviewers for their comments and suggestions.

Reviewer 1:
1) We apologize for omitting the Vertechi et al. reference and will cite it in the revision.
2) In the revision, we will add intuitive explanations of the objective functions.
3) In the revision, we will remove the argument for dimensionality reduction in a single neuron.
4) We are sorry that the term "signal-to-noise ratio" was confusing. By noise we mean dimensions in the input that are either non-informative or that carry little information and can be ignored without loss of performance. This will be clarified in the revision.
5) In our simulations we used randomly selected initial synaptic weights which seems biologically plausible.

Reviewer 2:
1) In the revision, we will clarify that unlike specifying a number of modes, specifying a threshold on eigenvalues allows changes in the number of output modes with changes in the stimulus statistics.
2) The reviewer is correct that our method does not necessarily reduce the number of output neurons. This allows adaptive changes to represented number of dimensions in the output. We will comment on this explicitly in the revision.
3) We agree with the reviewer that a relative threshold would be better. We will comment on this in the revision.
4) In the revision, we will remove the argument for dimensionality reduction in a single neuron.

Reviewer 3:
We appreciate the reviewer's thoughtful and fair concerns that the learning rules we get are specific to certain network architectures and cannot be generalized to other networks. While this is true, we believe that it is not a "bug" but a "feature" of our approach. For one, the same holds in the brain: different areas in the brain exhibit different plasticity rules, as exemplified in the plethora of STDP rules. Our work suggests, and perhaps this is the universal prediction the reviewer is looking for, that learning rules and network architectures are simultaneously specialized for solving a computation task and that there is a relation between them. We will make this point explicitly in the revised version.

Reviewer 5:
1) In general, we define the dimensionality of the output not as the number of active output neurons, but as the number of non-zero eigenvalues of the output covariance matrix. As such, our definition takes into account correlations in the neuronal output. Only in the case of decorrelated output units, resulting from the inclusion of extra terms in the objective Eq.(27), the dimensionality is the number of active output neurons. We will make this point more explicit to avoid confusions in the revision.
2) Regarding the time scale of neural activity in our networks and the retina, we focused on dimensionality reduction in the spatial domain leaving the temporal domain beyond the scope of this work. We will comment on this explicitly in the revision.
3) We will tone down our claim about normative theories of neural dimensionality reduction. However, we would like to point that our paper addresses a different problem than the literature on adaptation, in that we are not satisfied by suggesting a principle for the input/output mapping of a neural circuit but we also require that the neural implementation of such mapping should follow from such principle, including neural architectures, dynamics and plasticity rules.

Reviewer 6:
1) While it is true that our theoretical results rely on linear algebra, it is by no means trivial to obtain an algorithm from matrix manipulations that could be implemented in a neural network with local operations performed at synapses. This constraint disallows the use of some very basic matrix operations, like matrix inversion. As a proof that there is merit in our work, we can point to reference [22], which used similar linear algebra tools as our paper, was granted a journal publication and coincidentally was communicated by Oja himself.
2) The algorithm that we obtain is not a trivial modification to Oja's rule, as Oja's rule if defined for a single neuron rather than a network. There is a long history of work trying to extend Oja's rule to networks. Yet, our work is novel in combining the rigor of derivation from principled objective functions with biological plausibility similar to Ref.[22] communicated by Oja, suggesting that he did not perceive this a trivial modification.
3) We would like to present comparisons to other algorithms, but we are unaware of other neural network algorithms that implement the three computations (soft-thresholding, hard-thresholding and whitening in the principal subspace) we present in the paper. However, in Ref.[22] a simplified version of our soft-thresholding network (threshold was at zero) was shown to perform well compared to other neural networks. We will cite these comparisons in the revision.